# CoMat: Aligning Text-to-Image Diffusion Model with Image-to-Text Concept Matching

**Dongzhi Jiang**[1], **Guanglu Song**[2], **Xiaoshi Wu**[1], **Renrui Zhang**[1,3], **Dazhong Shen**[3],
**Zhuofan Zong**[1,2], **Yu Liu**[2✉], **Hongsheng Li**[1,3,4✉]

[1]CUHK MMLab, [2]SenseTime Research, [3]Shanghai AI Laboratory, [4]CPII under InnoHK
{dzjiang, wuxiaoshi, zhangrenrui}@link.cuhk.edu.hk   songguanglu@sensetime.com
{dazh.shen, zongzhuofan, liuyuisanai}@gmail.com   hsli@ee.cuhk.edu.hk

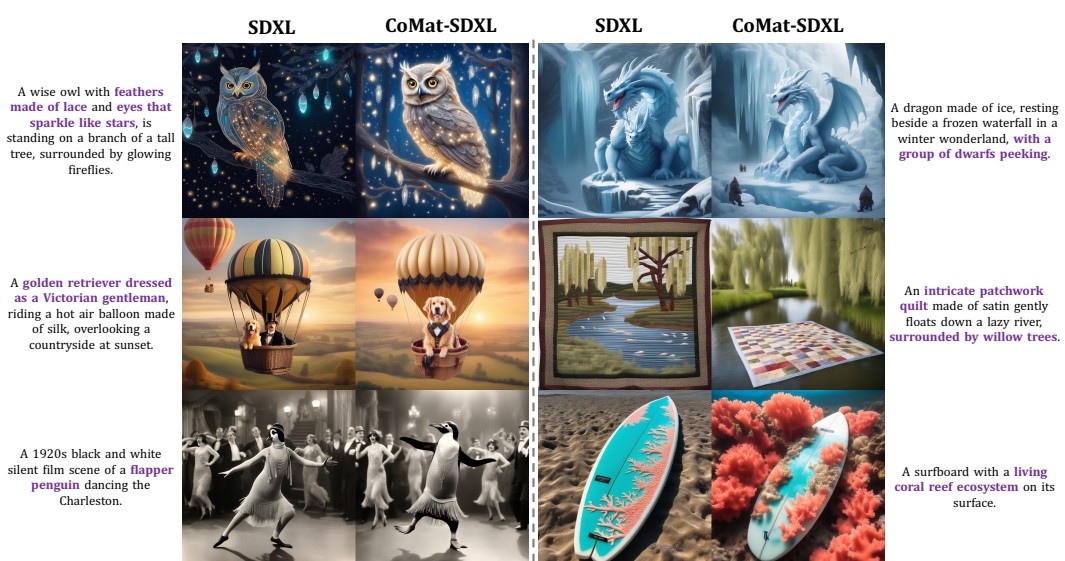

Figure 1: Current text-to-image diffusion model still struggles to produce images well-aligned with text prompts, as shown in the generated images of SDXL [49]. Our proposed method, CoMat, significantly enhances the baseline model on text condition following, demonstrating superior capability in text-image alignment. All the pairs are generated with the same random seed.

## Abstract

Diffusion models have demonstrated great success in the field of text-to-image generation. However, alleviating the misalignment between the text prompts and images is still challenging. We break down the problem into two causes: concept ignorance and concept mismapping. To tackle the two challenges, we propose CoMat, an end-to-end diffusion model fine-tuning strategy with the image-to-text concept matching mechanism. Firstly, we introduce a novel image-to-text concept activation module to guide the diffusion model in revisiting ignored concepts. Additionally, an attribute concentration module is proposed to map the text conditions of each entity to its corresponding image area correctly. Extensive experimental evaluations, conducted across three distinct text-to-image alignment benchmarks, demonstrate the superior efficacy of our proposed method, CoMat-SDXL, over the baseline model, SDXL [49]. We also show that our method enhances general condition utilization capability and generalizes to the long and complex prompt despite not specifically training on it. The code is available at https://github.com/CaraJ7/CoMat.

38th Conference on Neural Information Processing Systems (NeurIPS 2024).

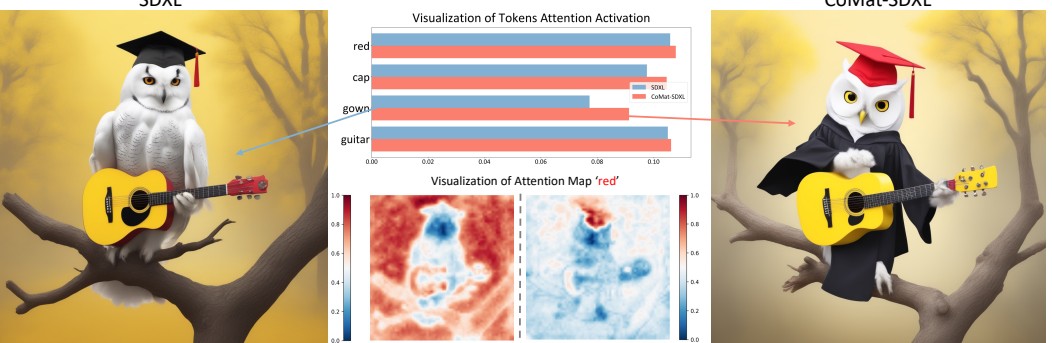

Figure 2: **Visualization of token activation and attention map**. We compare the tokens' attention activation value and attention map before and after applying our methods. Our method improves token activation and encourages the missing concept 'gown' to appear. Furthermore, the attention map of the attribute token 'red' better aligns with its region in the image.

# 1 Introduction

The area of text-to-image generation has witnessed considerable progress with the introduction of diffusion models [23, 51, 52, 56, 59, 76, 86] recently. These models have demonstrated remarkable performance in creating high-fidelity and diverse images based on textual prompts. However, it still remains challenging for these models to faithfully align with the prompts, especially for the complex ones. For example, as shown in Fig. 1, current state-of-the-art open-sourced model SDXL [49] fails to generate entities or attributes mentioned in the prompts, e.g., *the feathers made of lace* and *dwarfs* in the top row. Additionally, it fails to understand the relationship in the prompt. In the middle row of Fig. 1, it mistakenly generates a *Victorian gentleman* and a *quilt* with a river on it.

We break down this misalignment problem into two causes: concept ignorance and concept mismapping. The concept ignorance problem is caused by the diffusion model's omission of certain concepts in the text prompt. Even though the concept token is activated, the diffusion model often fails to map it to the correct area in the image, which is termed the concept mismapping problem. Actually, the misalignment originally stems from the training paradigm of the text-to-image diffusion models: Given the text condition $c$ and the paired image $x$, the training process aims to learn the conditional distribution $p(x|c)$. However, the text condition only serves as additional information for the denoising loss. Without explicit guidance in learning each concept in the text, the diffusion model could easily fail to understand the concepts in the prompt correctly.

Recently, to alleviate the misalignment, various works have proposed to incorporate linguistics prior [53, 6] to heuristically address the concept omission or concept mismapping problem. However, a specific design is required for each type of misalignment problem. Other works use the Large Language Model (LLM) [41, 77] to split the prompt into single entities and generate each of them. Although this method promotes the congruence between the image's global structure and the text prompt, it still suffers from local misalignment of the single entity. Hence, we ask the question: ***Is there a universal solution to address various global and local misalignment problems?***

In this work, we propose CoMat, an end-to-end fine-tuning strategy to enhance the prompt understanding and following by a novel image-to-text matching mechanism. **Concept Activation** module is proposed to address the concept ignorance problem. Given the generated image $\hat{x}$ conditioning on the prompt $c$, we seek to model and maximize the posterior probability $p(c|\hat{x})$ using a pre-trained image-to-text model. In contrast to regarding the textual prompt merely as a condition, as performed in the pre-training phase of the diffusion model, our approach incorporates the condition as a supervisory signal during the training process. Thanks to the proficiency of the image-to-text model in concept matching, whenever a particular concept is absent from the generated image, the diffusion model is steered to incorporate it within the image generation process. The guidance forces the diffusion model to revisit the ignored conditions and attend more to them. As an illustrative example shown in Fig. 2, the ignored concept in the image (e.g., the gown) possesses low attention activation values. After applying our method, we observe increased attention activation of each key concept,

contributing to the aligned image. In addition, considering the catastrophic forgetting issue arising from the new optimization objective, we also introduce a novel fidelity preservation module and mixed latent strategy to preserve the generation capability of the diffusion model. As for the concept mismapping problem, we find it especially prevails among the attributes of the objects. Hence, the **Attribute Concentration** module is introduced to promote both positive and negative mapping. We match the concept of attribute tokens in the text prompt to the generated image, with the insight that the attribute tokens should only be activated within its entity's area. Since the concept is a general term for a variety of features, our method can address both global structures and local details.

As an end-to-end method, no extra overhead is introduced during inference. We also show that our method is composable with methods leveraging external knowledge. Our contributions are summarized as follows:

- We propose CoMat, a text-to-image diffusion model fine-tuning strategy to effectively enhance the condition utilization capability by explicitly addressing the condition ignorance and incorrect condition mapping problem.
- We introduce the concept activation module equipped with fidelity preservation and mixed latent strategy to facilitate concept generation and attribute concentration module to foster correct concept mapping from text to image.
- Extensive quantitative and qualitative comparisons with baseline models indicate that our method significantly improves the text-image alignment in various scenarios, including object existence, attribute binding, relationship, and complex prompts.

## 2 Related Work

Recently, text-to-image diffusion models [56, 51, 65, 66] have become extremely trending, but they have also brought many new challenges [28, 61]. Among them, the text-to-image alignment problem has gained much attention. The problem is defined as the incoherence between the prompts and the generated images, which involves multiple aspects including existence, attribute binding, relationship, etc. Recent methods address the problem mainly in three ways.

Attention-based methods [6, 53, 47, 70, 2, 40] aim to modify or add restrictions on the attention map in the attention module in the UNet. This type of method often requires a heuristic design for each misalignment problem.

Planning-based methods first obtain the image layouts, either from the input of the user [39, 11, 32, 74, 15] or the generation of the Large Language Models (LLM) [48, 77, 69], and then produce aligned images conditioned on the layout. In addition, a few works propose to further refine the image with other vision expert models [55, 72, 71, 77]. Although such method splits a compositional prompt into single objects, it does not resolve the inaccuracy of the downstream diffusion model and still suffers from incorrect attribute binding problems. Besides, it exerts nonnegligible costs during inference.

Moreover, some works aim to enhance the alignment using feedback from image understanding models. [28, 62] fine-tune the diffusion model with well-aligned generated images chosen by the VQA model [36] to strategically bias the generation distribution. Other works propose to optimize the diffusion models in an online manner. [17, 4] introduce RL fine-tuning for generic rewards. As for differentiable reward, [14, 75, 73] propose to backpropagate the reward function gradient through the denoising process. Other works like [46] enhance the prompt encoding to foster better alignment. Similar to our work, [18] also proposes to leverage image captioning models. We discuss the difference between their method with ours in Appendix C.

## 3 Preliminaries

We implement our method on the leading text-to-image diffusion model, Stable Diffusion [56], which belongs to the family of latent diffusion models (LDM). In the training process, a normally distributed noise $\epsilon$ is added to the original latent code $z_0$ with a variable extent based on a timestep $t$ sampling from $\{1, ..., T\}$. Then, a denoising function $\epsilon_\theta$, parameterized by a UNet backbone, is trained to predict the noise added to $z_0$ with the text prompt $\mathcal{P}$ and the current latent $z_t$ as the input. Specifically, the text prompt is first encoded by the CLIP [50] text encoder $W$, then incorporated into the denoising

| SDXL | Playground v2 | PixArt-Alpha | CoMat-SDXL | |
|---|---|---|---|---|

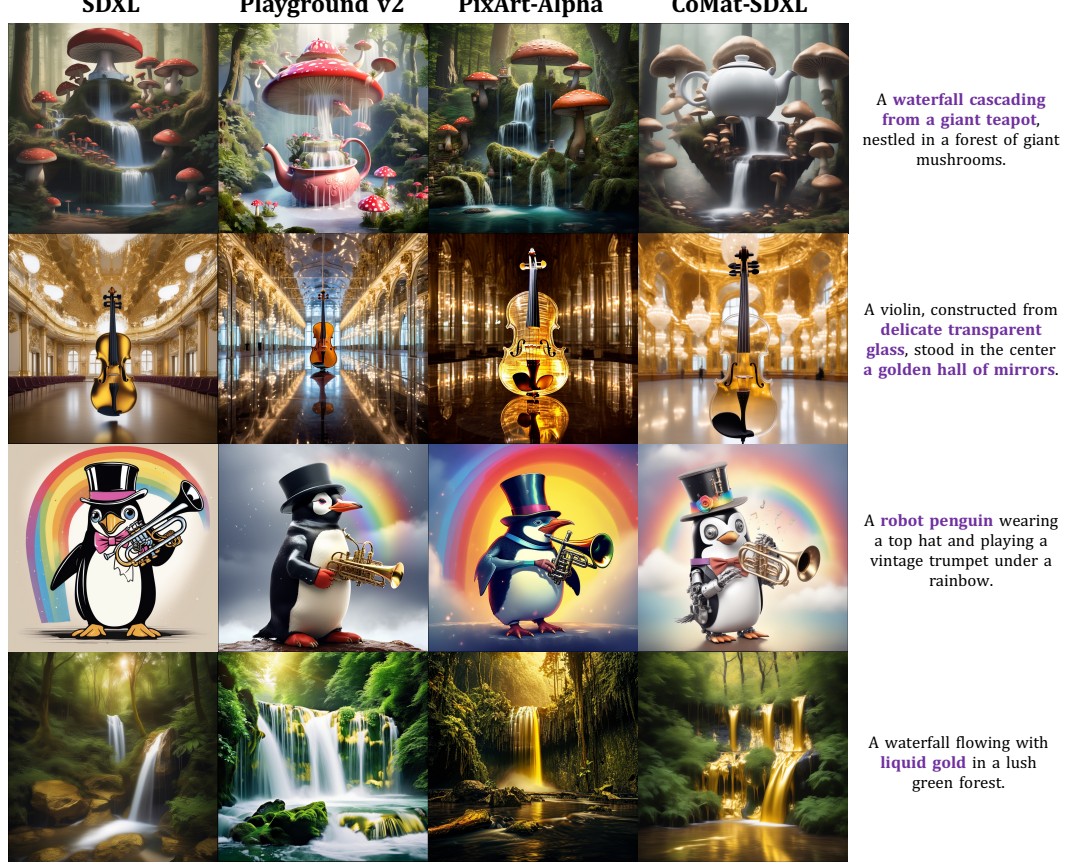

A **waterfall cascading from a giant teapot**, nestled in a forest of giant mushrooms.

A violin, constructed from **delicate transparent glass**, stood in the center **a golden hall of mirrors**.

A **robot penguin** wearing a top hat and playing a vintage trumpet under a rainbow.

A waterfall flowing with **liquid gold** in a lush green forest.

Figure 3: We showcase the results of our CoMat-SDXL compared with other state-of-the-art models. CoMat-SDXL consistently generates more faithful images.

function $\epsilon_\theta$ by the cross-attention mechanism. Concretely, for each cross-attention layer, the latent and text embedding is linearly projected to query $Q$ and key $K$, respectively. The cross-attention map $A^{(i)} \in \mathbb{R}^{h \times w \times l}$ is calculated as $A^{(i)} = \text{Softmax}(\frac{Q^{(i)}(K^{(i)})^T}{\sqrt{d}})$, where $i$ is the index of head. $h$ and $w$ are the resolution of the latent, $l$ is the token length for the text embedding, and $d$ is the feature dimension. $A_{i,j}^n$ denotes the attention score of the token index $n$ at the position $(i,j)$. The denoising loss in diffusion models' training is formally expressed as:

$$\mathcal{L}_{\text{LDM}} = \mathbb{E}_{z_0,t,p,\epsilon \sim \mathcal{N}(0,I)} \left[ \|\epsilon - \epsilon_\theta (z_t, t, W(\mathcal{P}))\|^2 \right]. \tag{1}$$

For inference, one draws a noise sample $z_T \sim \mathcal{N}(0, I)$, and then iteratively uses $\epsilon_\theta$ to estimate the noise and compute the next latent sample.

## 4 Method

The overall framework of our method is shown in Fig. 4. In Section 4.1, we first illustrate the concept activation module. Following this, we detail how we maintain the generation capability of the diffusion model by the fidelity preservation module and mixed latent strategy. Subsequently, in Section 4.2, we introduce the attribute concentration module for promoting attribute binding, and then we integrate the two components for joint learning.

### 4.1 Concept Activation

As noted in Section 1, the diffusion model occasionally exhibits little attention on certain concepts, and the corresponding concept is therefore missing in the image, which we termed as the condition

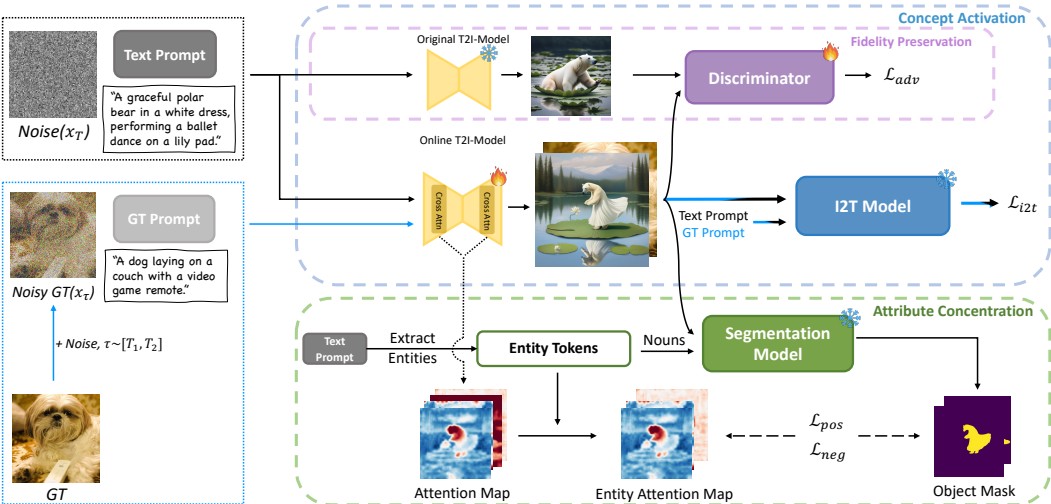

Figure 4: **Overview of CoMat**. The text-to-image diffusion model (T2I-Model) first generates an image according to the text prompt. Then the image is sent to the concept activation module and attribute concentration module to compute the loss for fine-tuning the online T2I-Model.

ignorance problem. To address this, our key insight is to add supervision on the generated image to detect the missing concepts. We achieve this by leveraging the image understanding ability of an image-to-text model, which can accurately identify concepts not present in the generated image based on the given text prompt. With the image-to-text model's supervision, the diffusion model is compelled to revisit text tokens to search for ignored condition information and assign more significance to the previously overlooked text concepts for better text-image alignment. Concretely, given a prompt $\mathcal{P}$ with word tokens $\{w_1, w_2, \ldots, w_L\}$, we first generate an image $\mathcal{I}$ with the denoising function $\epsilon_\theta$ after $T$ denoising steps. Then, a frozen image-to-text model $\mathcal{C}$ is used to score the alignment between the prompt and the image in the form of log-likelihood. The scoring capability of $\mathcal{C}$ comes with the image-to-text models' training nature. These models are trained for the image captioning task with the negative loglikelihood loss, i.e., the model needs to maximize the probability of generating the caption given the corresponding image. Therefore, whenever the generated image does not align with the text prompt, the model will output a low log-likelihood. Our training objective aims to minimize the negative of the log-likelihood, denoted as $\mathcal{L}_{i2t}$:

$$\mathcal{L}_{i2t} = -\log(p_\mathcal{C}(\mathcal{P}|\mathcal{I}(\mathcal{P}; \epsilon_\theta))) = -\sum_{i=1}^{L} \log(p_\mathcal{C}(w_i|\mathcal{I}, w_{1:i-1})). \tag{2}$$

Besides, it is also important to note that the concepts in the image include a broad field. This method provides a universal solution to various misalignment problems like object existence, complex relationships, etc. Finally, to conduct the gradient update through the whole iterative denoising process, we follow [73] to fine-tune the denoising network $\epsilon_\theta$, which ensures the training effectiveness and efficiency by simply stopping the gradient of the denoising network input.

However, since this fine-tuning process is purely piloted by the knowledge from the image-to-text model, the diffusion model could quickly overfit to the image-to-text model, lose its original capability, and produce deteriorated images, as shown in Fig. 7. To address this hacking issue, we introduce a novel fidelity preservation module and a mixed latent training strategy to preserve the generation ability of the diffusion model and guide the learning process.

**Fidelity Preservation.** We propose a novel adversarial loss that uses a discriminator to differentiate between images generated by pre-trained and fine-tuned diffusion models. Instead of using real-world images as the real data input for the discriminator, we use images generated by the original pre-trained diffusion model. This choice is based on the significant gap that still exists between the images generated by the original diffusion model and real-world images. Simply aligning the distribution of images generated by the fine-tuned diffusion model with that of real-world images would pose an undesired challenge for the learning process. For the discriminator $\mathcal{D}_\phi$, we initialize it with the pre-trained UNet in the Stable Diffusion model. The choice is motivated by the fact that the

pre-trained UNet shares similar knowledge with the online training model and fits well with the input domain. In our practice, this also enables the adversarial loss to be directly calculated in the latent space instead of the image space. Concretely, given a single text prompt, we employ the original diffusion model and the online training model to respectively generate image latent $\hat{z}_0$ and $\hat{z}'_0$. The adversarial loss is then computed as follows:

$$\mathcal{L}_{adv} = \log\left(\mathcal{D}_\phi\left(\hat{z}_0\right)\right) + \log\left(1 - \mathcal{D}_\phi\left(\hat{z}'_0\right)\right). \tag{3}$$

We aim to fine-tune the online model to minimize this adversarial loss, while concurrently training the discriminator to maximize it.

**Mixed Latent Strategy.** Besides, we inject information from real-world images to guide the learning process. Specifically, in addition to the latents starting from pure noise (marked as black in Fig. 4), we obtain the noisy real latents by adding noise on a real-world image at a random timestep $\tau$ (marked as 'Noisy GT'). We jointly denoise these two types of latents and calculate the loss given by the image-to-text model. The intuition is that, since the noisy real latent is a perturbed version of the real-world image, which is well aligned with its prompt, this provides a shortcut for the diffusion model to directly reconstruct the original image. This guidance can not only smooth the optimization process, but also prohibits the gradient from simply hacking the image-to-text model and encourages the diffusion model to generate an image both aligned with the prompt and of high fidelity. More illustration is included in Appendix A.

## 4.2 Attribute Concentration

Except for paying enough attention to the concept, the diffusion model must also map the concepts correctly on the image. As we dive into the generation process by visualizing the token attention activation map, we find that, for the attribute token, even though it is activated, it fails to attend to the correct area in the image and still causes the misalignment, e.g., 'yellow' in Fig. 5. Hence, we introduce the attribute concentration module to encourage the positive and discourage the negative concept mapping of attributes.

Specifically, we first extract all the entities $\{e_1, .., e_N\}$ in the prompts. An entity can be defined as a tuple of a noun $n_i$ and its attributes $a_i$, i.e., $e_i = (n_i, a_i)$, where both $n_i$ and $a_i$ are the sets of one or multiple tokens. We employ spaCy's transformer-based dependency parser [24] to parse the prompt to find all entity nouns, and then collect all attributes for each noun. A predefined set of nouns is established for filtering, including nouns that are abstract (e.g., scene, atmosphere, language), difficult to identify their area (e.g., sunlight, noise, place), or describe the background (e.g., morning, bathroom, party). Given all the selected nouns, we use them to prompt an open vocabulary segmentation model, Grounded-SAM [55], to find their corresponding regions as a binary mask $\{M^1, ..., M^N\}$. It is worth emphasizing that, to guarantee the segmentation accuracy, we only use the nouns of entities, excluding their associated attributes, as prompts for segmentation, considering the diffusion model could likely ignore the attribute or assign a wrong one to the object. Taking the 'suitcase' object in Fig 5 as an example, the model ignored the 'purple' attribute. Consequently, if the prompt 'purple suitcase' is given to the segmentor, it will fail to identify the entity's region. These inaccuracies can lead to a cascade of errors in the following process.

We add supervision to promote the diffusion model to map the entity tokens to the positive area, i.e., the entity area, and not to attend to the negative area, i.e., the other area:

$$\mathcal{L}_{\text{pos}} = -\frac{1}{N}\sum_{i=1}^{N}\sum_{M_{u,v}^i=1}\left(\alpha\sum_{k\in n_i\cup a_i}\left(\frac{A_{u,v}^k}{\sum_{x,y}A_{x,y}^k}\right) + \beta\frac{\log(\sum_{k\in n_i\cup a_i}A_{u,v}^k)}{|A|}\right), \tag{4}$$

$$\mathcal{L}_{\text{neg}} = -\frac{1}{N}\sum_{i=1}^{N}\sum_{M_{u,v}^i=0}\left(\alpha\underbrace{\sum_{k\in n_i\cup a_i}\left(\frac{-A_{u,v}^k}{\sum_{x,y}A_{x,y}^k}\right)}_{\text{region-level}} + \beta\underbrace{\frac{\log(1-\sum_{k\in n_i\cup a_i}A_{u,v}^k)}{|A|}}_{\text{pixel-level}}\right), \tag{5}$$

where $|A|$ is the number of pixels on the attention map, $\alpha$ and $\beta$ are two scaling factors, and $M^i$ should be resized to the resolution for each attention map $A$. The loss function covers the level of

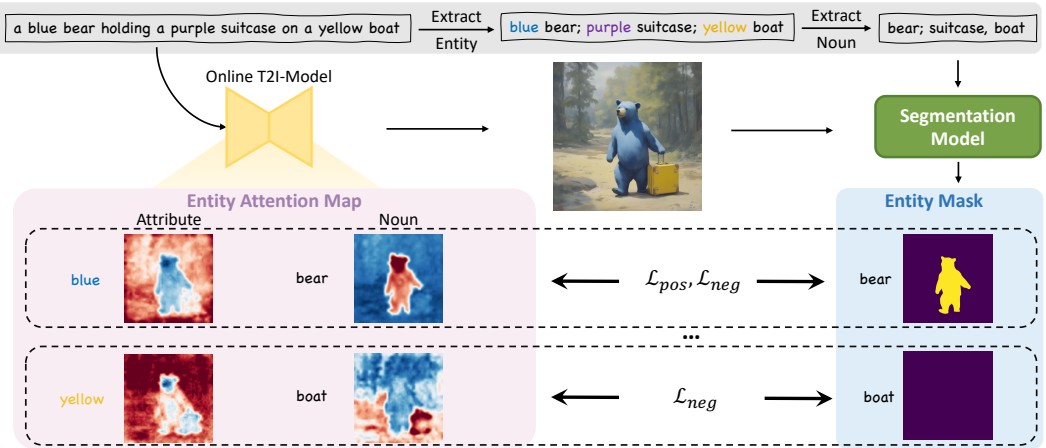

Figure 5: **Overview of Attribute Concentration**. Given a prompt, we first generate an image and record the cross-attention map for each token. We then identify regions of each entities in the prompt using the segmentation model. Finally, we optimize for the consistency between the entity attention map and its respective area in the image by encouraging positive and discouraging negative mapping.

regions and pixels. Take the $\mathcal{L}_{\text{pos}}$ for example. We restrict the attention of each entity tokens $e_i$ only activated inside the positive region by the region-level loss. We further restrict each pixel in the positive region to only attend to entity tokens by the pixel-level loss. We take into account the scenario where certain objects in the prompt do not appear in the generated image due to misalignment. In this case, the negative loss of pixels is still valid. When the mask is entirely zero, it signifies that none of the pixels should attend to the missing entity tokens in the current image.

Finally, we combine the image-to-text model loss, adversarial loss and attribute concentration loss to build up our training objectives for the online diffusion model as follows:

$$\mathcal{L} = \mathcal{L}_{\text{i2t}} + \mathcal{L}_{\text{pos}} + \mathcal{L}_{\text{neg}} + \lambda \mathcal{L}_{\text{adv}}, \tag{6}$$

where $\lambda$ are scaling factors to balance the loss. We provide the pseudocode for the loss computation process in Algorithm 1.

## 5 Experiment

### 5.1 Experimental Setup

Table 1: T2I-CompBench result. The best score is in blue , with the second-best score in green .

| Model | Attribute Binding | | | Object Relationship | | Complex↑ |
|---|---|---|---|---|---|---|
| | Color ↑ | Shape↑ | Texture↑ | Spatial↑ | Non-Spatial↑ | |
| StructureDiffusion [19] | 0.4990 | 0.4218 | 0.4900 | 0.1386 | 0.3111 | 0.3355 |
| Composable Diffusion [44] | 0.4063 | 0.3299 | 0.3645 | 0.0800 | 0.2980 | 0.2898 |
| Attend-and-Excite [6] | 0.6400 | 0.4517 | 0.5963 | 0.1455 | 0.3109 | 0.3401 |
| TokenCompose [70] | 0.5055 | 0.4852 | 0.5881 | 0.1815 | 0.3173 | 0.2937 |
| PixArt-$\alpha$ [7] | 0.6690 | 0.4927 | 0.6477 | 0.2064 | 0.3197 | 0.3433 |
| Playground-v2 [33] | 0.6208 | 0.5087 | 0.6125 | 0.2372 | 0.3098 | 0.3613 |
| SD1.5 [56] | 0.3758 | 0.3713 | 0.4186 | 0.1165 | 0.3112 | 0.3047 |
| **CoMat-SD1.5 (Ours)** | 0.6734 | 0.5064 | 0.6243 | 0.2073 | 0.3166 | 0.3575 |
| | *(+0.2976)* | *(+0.1351)* | *(+0.2057)* | *(+0.0908)* | *(+0.0054)* | *(+0.0528)* |
| SDXL [49] | 0.5879 | 0.4687 | 0.5299 | 0.2131 | 0.3119 | 0.3237 |
| **CoMat-SDXL (Ours)** | 0.7827 | 0.5329 | 0.6468 | 0.2428 | 0.3187 | 0.3680 |
| | *(+0.1948)* | *(+0.0642)* | *(+0.1169)* | *(+0.0297)* | *(+0.0068)* | *(+0.0443)* |

**Base Model Settings.** We mainly implement our method on SDXL [56] for all experiments, and we also evaluate our method on Stable Diffusion v1.5 [56] (SD1.5). For the captioning model, we choose

BLIP [36] fine-tuned on COCO [42] image-caption data. We adopt the pre-trained UNet of SD1.5 as the discriminator in the fidelity preservation module. More training details are in Appendix E.1.

**Dataset.** Since the prompt to the diffusion model needs to be challenging enough to lead to missing concepts, we directly utilize the training data or text prompts provided in existing text-to-image alignment benchmarks. Specifically, the training data includes the training set provided in T2I-CompBench [28], all the data from HRS-Bench [3], and 5,000 prompts randomly chosen from ABC-6K [20]. Altogether, these amount to around 20,000 text prompts. Note that the training set composition can be freely adjusted according to the ability targeted to improve. The text-image pairs used in the mixed latent strategy are from the training set of COCO [42].

**Benchmarks.** We evaluate our method on three text-image alignment benchmarks and follow their default settings. T2I-CompBench [28] comprises 6,000 compositional text prompts evaluating 3 categories (attribute binding, object relationships, and complex compositions) and 6 sub-categories (color binding, shape binding, texture binding, spatial relationships, non-spatial relationships, and complex compositions). TIFA [27] uses pre-generated question-answer pairs and a VQA model to evaluate the generation results with 4,000 diverse text prompts and 25,000 questions across 12 categories. DPG-Bench [26] composes 1065 dense prompts with an average token length of 83.91. The prompt depicts a much more complex scenario with diverse objects and adjectives.

Table 2: TIFA and DPG-Bench results.

| Model | TIFA↑ | DPG↑ |
|---|---|---|
| PixArt-$\alpha$ [7] | 82.9 | 71.11 |
| Playground-v2 [33] | 86.2 | 74.54 |
| SD1.5 [56] | 78.4 | 63.18 |
| **CoMat-SD1.5 (Ours)** | 85.8 | 73.32 |
| | *(+7.4)* | *(+10.14)* |
| SDXL [49] | 85.9 | 74.65 |
| **CoMat-SDXL (Ours)** | **87.5** | **77.13** |
| | *(+1.6)* | *(+2.48)* |

Table 3: FID-10K result.

| Model | $\mathcal{D}_\phi$ | $\mathcal{D}_\phi$ input | ML | **FID-10K↓** |
|---|---|---|---|---|
| SD1.5 [56] | N/A | N/A | N/A | 16.69 |
| CoMat-SD1.5 | ✗ | N/A | ✗ | 19.02 |
| CoMat-SD1.5 | UNet [57] | real-world latent | ✗ | 17.99 |
| CoMat-SD1.5 | UNet [57] | generated latent | ✗ | 16.69 |
| CoMat-SD1.5 | DINO [5] | generated image | ✗ | 23.86 |
| CoMat-SD1.5 | UNet [57] | generated latent | ✔ | **15.43** |

## 5.2 Quantitative Results

We compare our methods with our baseline models: SD1.5 and SDXL, and two state-of-the-art open-sourced text-to-image models: PixArt-$\alpha$ [7] and Playground-v2 [33].

**T2I-CompBench**. The evaluation result is shown in Table 1. Note that we cannot reproduce results reported in some relevant works [7, 28] due to the evolution of the evaluation code. All our shown results are based on the latest code released in GitHub[1]. We observe significant gains in all six sub-categories compared with our baseline models. With our methods, SD1.5 can even achieve better or comparable results compared with PixArt-$\alpha$ and Playground-v2. Our CoMat-SDXL demonstrates the best performance regarding attribute binding, spatial relationships, and complex compositions.

**TIFA.** We show the results in TIFA in Table 2. Our CoMat-SDXL achieves the best performance with an improvement of 1.6 scores compared to SDXL. Besides, CoMat significantly enhances SD1.5 by 7.4 scores, which largely surpasses PixArt-$\alpha$.

**DPG-Bench.** The results in DPG-Bench is shown in Table 2. Although we do not train our model on dense prompts and can only accept 77 tokens, similar to Stable Diffusion, our method successfully generalizes to this more complex scenario and brings significant improvement to the baseline model.

## 5.3 Qualitative Results

Fig. 3 presents a side-by-side comparison between CoMat-SDXL and other state-of-the-art diffusion models. We observe these models exhibit inferior condition utilization ability compared with CoMat-SDXL. Prompts in Fig. 3 all possess concepts that are contradictory to real-world phenomena. All the three compared models stick to the original bias and choose to ignore the unrealistic content (e.g., waterfall cascading from a teapot, transparent violin, robot penguin, and waterfall of liquid gold), which causes misalignment. However, by training to faithfully align with the conditions in the prompt, CoMat-SDXL follows the unrealistic conditions and provides well-aligned images. The user study result and more visualization result is detailed in Appendix B.1 and F.2.

---

[1] https://github.com/Karine-Huang/T2I-CompBench

Table 4: Impact of concept activation and attribute concentration. 'CA' and 'AC' denote concept activation and attribute concentration respectively.

| Model | CA | AC | Attribute Binding | | | Object Relationship | | Complex↑ |
|-------|----|----|-------------------|--|--|---------------------|--|----------|
| | | | Color ↑ | Shape↑ | Texture↑ | Spatial↑ | Non-Spatial↑ | |
| SDXL | | | 0.5879 | 0.4687 | 0.5299 | 0.2131 | 0.3119 | 0.3237 |
| SDXL | ✓ | | 0.7455 | 0.5043 | 0.6252 | 0.2321 | 0.3171 | 0.3660 |
| SDXL | ✓ | ✓ | **0.7827** | **0.5329** | **0.6468** | **0.2428** | **0.3187** | **0.3680** |

Table 5: The impact of different image-to-text models.

| Image-to-text Model | Attribute Binding | | | Object Relationship | | Complex↑ |
|---------------------|-------------------|--|--|---------------------|--|----------|
| | Color ↑ | Shape↑ | Texture↑ | Spatial↑ | Non-Spatial↑ | |
| BLIP [36] | **0.7827** | **0.5329** | **0.6468** | **0.2428** | **0.3187** | **0.3680** |
| GIT [67] | 0.6916 | 0.5146 | 0.5971 | 0.2404 | 0.3149 | 0.3413 |
| LLaVA [43] | 0.6338 | 0.4722 | 0.5518 | 0.1963 | 0.3117 | 0.3286 |
| N/A | 0.5879 | 0.4687 | 0.5299 | 0.2131 | 0.3119 | 0.3237 |

## 5.4 Ablation Study

**Effectiveness of Concept Activation and Attribute Concentration.** In Table 4, we show the T2I-CompBench result aiming to identify the effectiveness of the concept activation and attribute concentration modules. We find that the concept activation module accounts for major gains to the baseline model. On top of that, the attribute concentration module brings further improvement to all six sub-categories in T2I-CompBench. We show the qualitative effectiveness in Fig. 6.

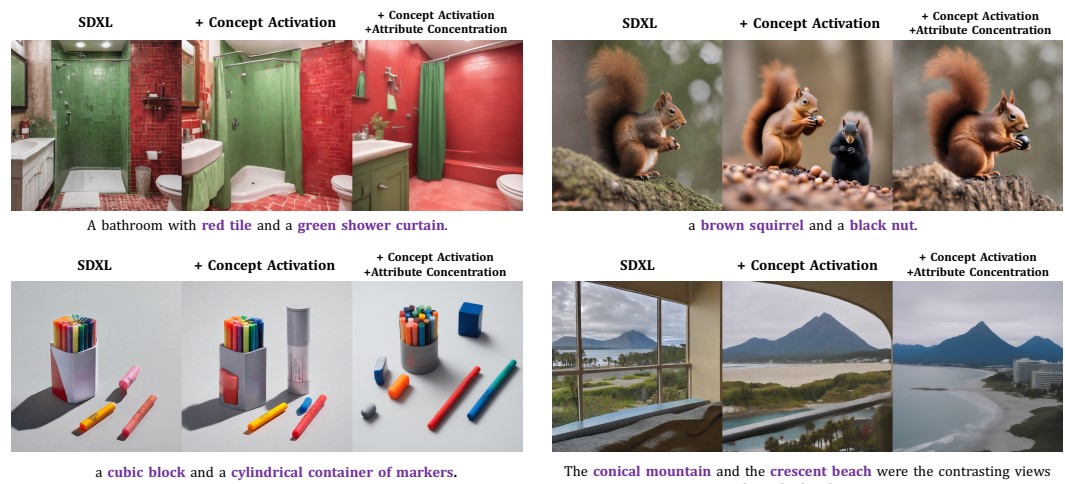

A bathroom with red tile and a green shower curtain.

a brown squirrel and a black nut.

a cubic block and a cylindrical container of markers.

The conical mountain and the crescent beach were the contrasting views from the hotel room.

Figure 6: Visualization of the effectiveness of the proposed modules. CA contributes to the existence of objects mentioned in the prompts. AC further guides the attention of the attributes to focus on their corresponding objects.

**Design of Fidelity Preservation and Mixed Latent.** We examine the photorealism of generated images to evaluate the generation capability. We calculate the FID [22] score using 10K data from the COCO validation set. As shown in Table 3 and Fig. 7, without any preservation method, the diffusion model only tries to hack the image-to-text model and loses its original generation ability with an increase of FID score from 16.69 to 19.02. Besides, inputting the latent generated by the original diffusion model performs better than the latent of real-world images. As for the discriminator architecture, the UNet is superior to a pre-trained DINO [5] which even interferes the training process. Finally, the Mixed Latent (ML) strategy further enhances the generated image quality.

| CoMat-SDXL (w/o FP&ML) | CoMat-SDXL | CoMat-SDXL (w/o FP&ML) | CoMat-SDXL |
| --- | --- | --- | --- |

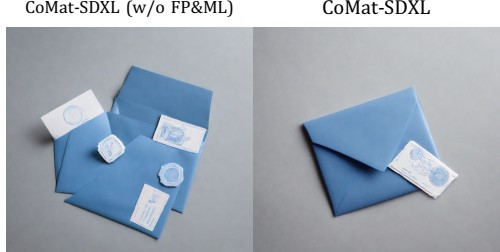 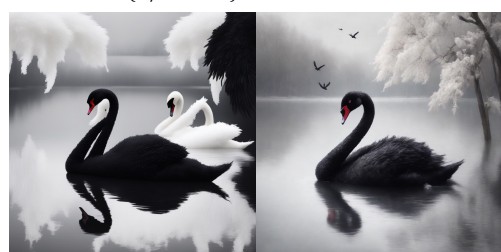

A blue envelop and a white stamp      A black swan and a white lake

Figure 7: Visualization result of the effectiveness of the Fidelity Preservation module (FP) and Mixed Latent (ML) strategy.

**Different Image-to-text Models.** We show the T2I-CompBench results with different image captioning models in Table 5. We find that all three image-to-text models can boost the performance of the diffusion model with our framework, where BLIP achieves the best performance. We provide more analysis on the choice of the image-to-text models in Appendix B.3.

## 5.5 Robustness Analysis

We test the robustness of our method by the method proposed in [16], which introduces an automated way to discover prompts that induce misalignment in Stable Diffusion models. We evaluate this attack method on SD1.5 and CoMat-SD1.5 using both short and long prompts.

For 1,000 ImageNet-1K classes, we generate 20 samples per class using the attack method and measure the success rate - defined as the proportion of generated images that could be mistakenly classified by a visual classifier. Table 6 shows that CoMat-SD1.5 exhibits lower attack success rates for both prompt lengths, demonstrating enhanced alignment robustness compared to the base model.

Table 6: Success rate of prompt attack.

| Model | Short prompt↓ | Long prompt↓ |
| --- | --- | --- |
| SD1.5 [56] | 49.1% | 51.1% |
| CoMat-SD1.5 | 46.8% | 50.4% |

## 6 Limitations

How to more effectively incorporate Multimodal Large Language Models (MLLMs) into text-to-image diffusion models by our proposed method requires more exploration. MLLM possesses state-of-the-art image-text understanding capability in addition to image captioning. We will focus on leveraging MLLMs to provide finer-grained guidance to the diffusion model in our future work. In addition, the attribute concentration module cannot assign attributes to multiple same-name objects, such as an Asian girl with an Indian girl, the segmentation model cannot differentiate two girls and therefore cannot assign attributes. As for the training cost, since our method needs the diffusion model to perform the whole inference process, the training time is extended. Our future direction will be to accelerate the training process.

## 7 Conclusion

In this paper, we propose CoMat, an end-to-end diffusion model fine-tuning strategy equipped with image-to-text concept matching. We identify the two causes of the misalignment problem and propose two key components to explicitly address them. The concept activation module leverages an image-to-text model to supervise the generated image and find out the ignored condition information. It also integrates the fidelity preservation module and mixed latent strategy to maintain the generation capability. Besides, we introduce the attribute concentration module to address the attribute mismapping issue. Through extensive experiments, we have demonstrated that CoMat largely outperforms its baseline model and even surpasses commercial products in multiple aspects. We hope our work can inspire future work on the cause of the misalignment and the solution to it.

## Acknoledgement

This project is funded in part by National Key R&D Program of China Project 2022ZD0161100, by the Centre for Perceptual and Interactive Intelligence (CPII) Ltd under the Innovation and Technology Commission (ITC)'s InnoHK, by General Research Fund of Hong Kong RGC Project 14204021. Hongsheng Li is a PI of CPII under the InnoHK.

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

# A Algorithm

Here we first detail the mixed latent training strategy and then provide the pseudocode for a single loss computation step.

The mixed latent strategy contains two types of latents in the fine-tuning procedure, i.e., the latent starting from the pure noise and the noisy latent from the GT Images.

**Latent starting from the pure noise** This serves as the main branch in our pipeline. Our fine-tuning process shares the same procedure to generate an image as the diffusion model does in the inference time. We uniformly sample $K$ steps from all the inference steps to enable the gradient. Therefore, the latent is sampled from the pure noise $\mathcal{N}(0, I)$. We iteratively denoise it to obtain the generated image. The image is then used to calculate the $\mathcal{L}_{i2t}$ and $\mathcal{L}_{adv}$ loss. It is also sent to the segmentation model to provide the object mask for computing the $\mathcal{L}_{pos}$ and $\mathcal{L}_{neg}$. The latent starting from the pure noise corresponds to the upper left part in Fig. 4. Please refer to [73] for how to receive the gradient from the loss.

**Noisy latent from the GT Images** We also aim to inject information from the GT images to stabilize the fine-tuning process. We randomly sample a timestamp $\tau$ from a pre-defined range $[T_1, T_2]$. Then we obtain $x_\tau$ by adding the timestamped noise $\epsilon_\tau$ on the latent of the GT Image $x_0$. We also iteratively denoise this noisy GT latent to get $\hat{x}_0$ as we do for the latents starting from the pure noise. This $\hat{x}_0$ is only used to calculate the $\mathcal{L}_{i2t}$ loss. The latent starting from the noisy GT corresponds to the bottom left part in Fig. 4.

The pseudocode for a single loss computation step for the online T2I-Model is described below.

---

**Algorithm 1** A single loss computation step for the online T2I-Model during fine-tuning

---

**Input**: Text prompt $\mathcal{P}$, GT Image $\mathcal{I}_g$, GT Prompt $\mathcal{P}_g$, original T2I-Model $\epsilon_{pre}$, online T2I-Model $\epsilon_\theta$, pre-trained I2T-Model $\mathcal{C}$, discriminator $\mathcal{D}_\phi$, segmentation model $\mathcal{S}$, timestep range $[T_1, T_2]$, timestep $\tau$, attention map $A$, scaler $\lambda$; $[;]$ denotes concatenate

1: $x_T, \xi \sim \mathcal{N}(0, I)$
2: $\tau \sim \text{Uniform}[T_1, T_2]$
3: $x_\tau = \text{AddNoise}(\mathcal{I}_g, \xi, \tau)$
4: $\hat{\mathcal{I}}, A = \text{GenerateImage}(\epsilon_\theta, x_T, \mathcal{P})$
5: $\hat{\mathcal{I}}_g, \_\_ = \text{GenerateImage}(\epsilon_\theta, x_\tau, \mathcal{P}_g)$
6: $\mathcal{L}_{i2t} = \text{ComputeI2TLoss}(\mathcal{C}, \left[\hat{\mathcal{I}}; \hat{\mathcal{I}}_g\right], [\mathcal{P}; \mathcal{P}_g])$
7: $\hat{\mathcal{I}}_{pre}, \_\_ = \text{GenerateImage}(\epsilon_{pre}, x_T, \mathcal{P})$
8: $\mathcal{L}_{adv} = \text{ComputeAdvLoss}(\mathcal{D}_\phi, \hat{\mathcal{I}}, \hat{\mathcal{I}}_{pre})$
9: $\mathcal{L}_{pos}, \mathcal{L}_{neg} = \text{ComputeAttrLoss}(\mathcal{S}, \hat{\mathcal{I}}, \mathcal{P}, A)$
10: $\mathcal{L} = \mathcal{L}_{i2t} + \mathcal{L}_{pos} + \mathcal{L}_{neg} + \lambda\mathcal{L}_{adv}$
**Output**: Training loss for the online T2I-Model $\mathcal{L}$

---

# B Additional Results and Analysis

## B.1 User preference study

We randomly select 100 prompts from DSG1K [13] and use them to generate images with SDXL [56] and our method (CoMat-SDXL). We ask 5 participants to assess both the image quality and text-image alignment. Human raters are asked to select the superior respectively from the given two synthesized images, one from SDXL, and another from our CoMat-SDXL. For fairness, we use the same random seed for generating both images. The voting results are summarised in Fig. 8. Our CoMat-SDXL greatly enhances the alignment between the prompt and the image without sacrificing the image quality.

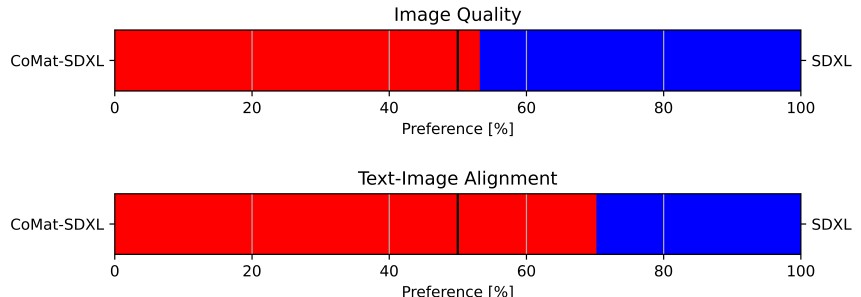

Figure 8: User preference study results.

## B.2 Composability with planning-based methods

Since our method is an end-to-end fine-tuning strategy, we demonstrate its flexibility in the integration with other planning-based methods, where combining our method also yields superior performance. RPG [77] is a planning-based method utilizing Large Language Model (LLM) to generate the description and subregion for each object in the prompt. We refer the reader to the original paper for details. We employ SDXL and our CoMat-SDXL as the base model used in [77] respectively. As shown in Fig. 9, even though the layout for the generated image is designed by LLM, SDXL still fails to faithfully generate the single object aligned with its description, e.g., the wrong mat color and the missing candle. Although the planning-based method generates the layout for each object, it is still bounded by the base model's condition following capability. Combining our method can therefore perfectly address this issue and further enhance alignment.

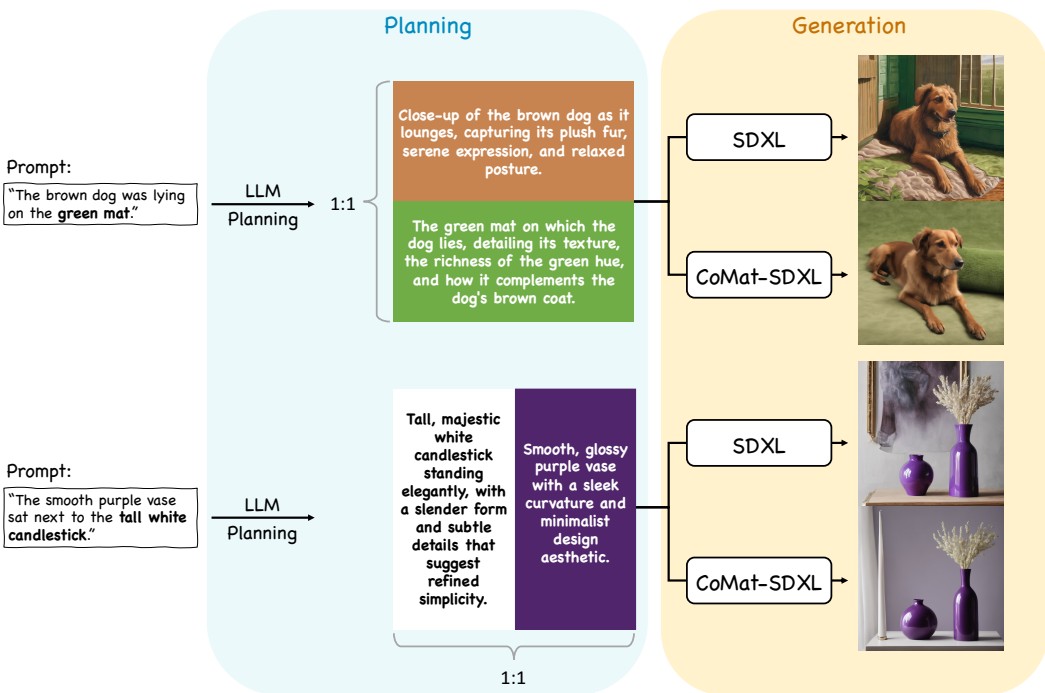

Figure 9: Pipeline for integrating CoMat-SDXL with planning-based method. CoMat-SDXL correctly generates the green mat in the upper row and the tall white candle in the bottom row.

### B.3 How to choose an image-to-text model?

We provide a further analysis of the varied performance improvements observed with different image-to-text models, as shown in Table 5 of the main text.

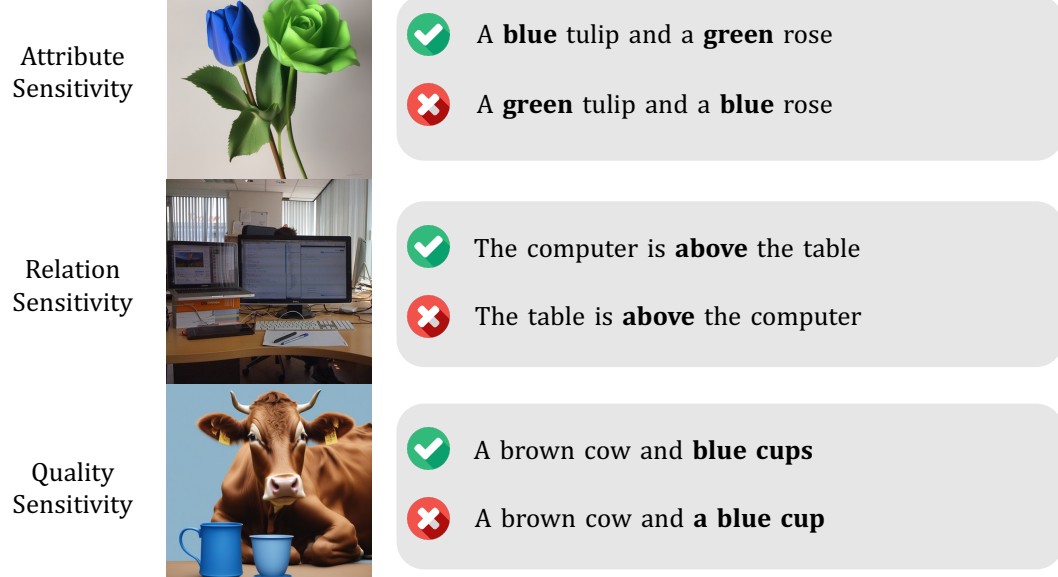

Figure 10: Examples for the three core sensitivities.

For an image-to-text model to be valid for the concept activation module, it should be able to tell whether each concept in the prompt appears and appears correctly. We construct a test set to evaluate this capability of the image-to-text model. Intuitively, given an image, a qualified image-to-text model should be sensitive enough to the prompts that faithfully describe it against those that are incorrect in certain aspects. We study three core demands for an image-to-text model:

- **Attribute sensitivity.** The image-to-text model should distinguish the noun and its corresponding attribute. The corrupted caption is constructed by switching the attributes of the two nouns in the prompt.
- **Relation sensitivity.** The image-to-text model should distinguish the subject and object of a relation. The corrupted caption is constructed by switching the subject and object.
- **Quantity sensitivity.** The image-to-text model should distinguish the quantity of an object. Here we only evaluate the model's ability to tell one from many. The corrupted caption is constructed by turning singular nouns into plural or otherwise.

We assume that they are the basic requirements for an image-to-text model model to provide valid guidance for the diffusion model. Besides, we also choose images from two domains: real-world images and synthetic images. For real-world images, we randomly sample 100 images from the ARO benchmark [79]. As for the synthetic images, we use the pre-trained SD1.5 [56] and SDXL [49] to generate 100 images according to the prompts in T2ICompBench [28]. These selections make up for the 200 images in our test data. We show the examples in Fig. 10.

For the sensitivity score, we compare the difference between the alignment score (i.e., log-likelihood) of the correct and corrupted captions for an image. Given the correct caption $\mathcal{P}$ and corrupted caption $\mathcal{P}'$ corresponding to image $\mathcal{I}$, we compute the sensitivity score $\mathcal{S}$ as follows:

$$\mathcal{S} = \frac{\log(p_{\mathcal{C}}(\mathcal{P}|\mathcal{I})) - \log(p_{\mathcal{C}}(\mathcal{P}'|\mathcal{I}))}{|\log(p_{\mathcal{C}}(\mathcal{P}|\mathcal{I}))|}. \tag{7}$$

Then we take the mean value of all the images in the test set. The result is shown in Table 7. The rank of the sensitivity score aligns with the rank of the gains brought by the image-to-text model model shown in the main text. Hence, except for the parameters, we argue that sensitivity is also a must for an image-to-text model to function in the concept activation module.

Table 7: Statistics of image-to-text models.

| Image-to-text Model | Parameters | Sensitivity Score |
|---|---|---|
| BLIP [36] | 469M | 0.1987 |
| GIT [67] | 394M | 0.1728 |
| LLaVA [43] | 7.2B | 0.1483 |

## C   More Related Work

The image-to-text model in the main text refers to the models capable of image captioning. Previous image captioning models are pre-trained on various vision and language tasks (e.g., image-text matching, (masked) language modeling) [37, 45, 31, 64], then fine-tuned with image captioning tasks [12]. Various model architectures have been proposed [68, 78, 35, 36, 67]. BLIP [36] takes a fused encoder architecture, while GIT [67] adopts a unified transformer architecture. Recently, multimodal large language models (MLLMs) have been flourishing [43, 85, 1, 87, 60, 30]. Empowered by the strong language ability of large language models (LLMs) [81], MLLMs are capable of various vision-language tasks like detailed image captioning [43, 85, 63, 8], visual question answering [38, 83, 84, 9, 29], etc. LLaVA [43, 34, 34] is one of the representative MLLMs. When prompted properly, it can generate elaborate image captions.

Similar to our work, [18] proposes to caption the generated images and optimize the coherence between the produced captions and text prompts. Although an image-to-text model is also involved, they fail to provide detailed guidance. It has been shown that the generated captions are prone to omit key concepts and involve undesired added features [27]. Besides, the method leverages a pre-trained text encoder to compute the similarity between the prompt and generated caption, which further causes information to be missed during text encoding. All these designs lead the optimization target to be vague and sub-optimal.

### C.1   vs. Differentiable Reward Method

**Similarity:** Our method is inspired by the technique introduced in the differentiable reward method to perform gradient update.

**Difference:** (1) **Reward Model.** Our method is the first to leverage an image-to-text model to perform image captioning on the generated image and compute the loss on the caption. (2) **No fidelity preservation.** The current differentiable reward method ignores the aspect of preserving the generation capability if not training against a reward of image quality. Our method introduces a novel fidelity preservation module, which utilizes a discriminator with similar knowledge to preserve the generation capability. This greatly alleviates the reward hacking problem introduced by only training with the differentiable reward method. (3) **No guidance from real-world image.** The current differentiable reward method all starts from pure noise. Since our method is optimizing for alignment, we can incorporate real-world image-text pairs to guide the optimization process. With our mixed latent strategy, the latent starting from the noise is conditioned on the difficult prompt to promote alignment, while the latent starting from the noisy GT image is used to prohibit the diffusion model from overfitting to the image-to-text model.

### C.2   vs. TokenCompose [70]

**Similarity:** Both [70] and our method incorporates the object mask to guide the attention of the diffusion model.

**Difference:** (1) **Limited and inferior optimizing target.** [70] merely focuses on optimizing the consistency between the noun mask and the object mask. However, as shown in Fig. 5, the attention mask of the noun (the 'bear' token) has already aligned well with the object mask. Optimizing for this consistency is inferior. On the other hand, our method focuses on a much broader area, i.e., entity tokens, which consist of nouns and their various associated attributes. We also find that the consistency between the attributes and the object mask bears very little similarity, which should be paid more attention. (2) **No negative concept mapping.** Since the training data of [70] is the real-world image-text pairs, all the nouns in the prompt show up in the image. However, this prohibits

the model from learning in a negative way, i.e., if the entity is not on the image, none of the pixel should be activated by this token. Our method leverages images generated by the diffusion model. The entity missing is common. The model obtains the chance to learn in a negative way. (3) **No difficult training data.** Another issue caused by training with image-text pairs is that the training data may be of a common scenario, which is easier to learn. Since our method does not need real-world images and only starts from the noise and text prompt, this enables a more efficient training process.

### C.3 vs. Class-specific Prior Preservation Loss [58]

**Similarity:** Both the class-specific prior preservation loss (CPP Loss) [58] and our proposed fidelity preservation module (FP) share the similar high-level idea of preserving the generation quality while fine-tuning the diffusion models.

**Difference:** (1) **Target task and preserve domain.** [58] seeks to personalize image generation for specific objects. While the introduced CPP Loss primarily maintains generative capabilities within a narrow domain—specifically, the object class present in the training data—our proposed FP module operates within the context of text-image alignment. FP aims to preserve general generative capabilities by computing adversarial loss across the entire training dataset, encompassing a diverse range of text prompts. (2) **Methodology.** Since the training data of [58] finetunes the diffusion model with the pretraining loss, i.e., the squared error denoising loss on a certain timestamp. CPP Loss follows its form. In contrast, our fine-tuning procedure simulates the inference process of the diffusion model to conduct a full-step inference. We aim to directly supervise the generated image to achieve the training-test alignment. Therefore, we propose the novel FP module to leverage a discriminator to adversarially preserve its quality. The applied discriminator is also updated along with the fine-tuning process, enabling finer control of the image quality.

## D Future Work

We believe our work can also be applied in the text-to-video diffusion models. With the introduction of various MLLMs handling videos [80, 10] and video segmentation models [54, 21], both our concept activation and attribute concentration modules could be used for text-video-alignment training.

# E  Experimental Setup

## E.1  Implementation Details

**Training Details.** In our method, we inject LoRA [25] layers into the UNet of the online training model and discriminator and keep all other components frozen. For both SDXL and SD1.5, we train 2,000 iters on 8 NVIDIA A100 GPUS. We use a local batch size of 6 for SDXL and 4 for SD1.5. We choose Grounded-SAM [55] from other open-vocabulary segmentation models [82, 88]. The DDPM [23] sampler with 50 steps is used to generate the image for both the online training model and the original model. In particular, we follow [73] and only enable gradients in 5 steps out of those 50 steps, where the attribute concentration module would also be operated. Besides, to speed up training, we use training prompts to generate and save the generated latents of the pre-trained model in advance, which are later input to the discriminator during fine-tuning.

**Training Resolutions.** We observe that training SDXL is very slow due to the large memory overhead at $1024 \times 1024$. However, SDXL is known to generate low-quality images at resolution $512 \times 512$. This largely affects the image understanding of the image-to-text model. So we first equip the training model with better image generation capability at $512 \times 512$. We use our training prompts to generate $1024 \times 1024$ images with pre-trained SDXL. Then we resize these images to $512 \times 512$ and use them to fine-tune the UNet of SDXL for 100 steps, after which the model can already generate high-quality $512 \times 512$ images. We continue to implement our method on the fine-tuned UNet.

**Training Layers for Attribute Concentration.** Following [70], only cross-attention maps in the middle blocks and decoder blocks are used to compute the loss.

**Hyperparameters Settings.** We provide the detailed training hyperparameters in Table 8.

Table 8: CoMat training hyperparameters for SD1.5 and SDXL.

| Name | SD1.5 | SDXL |
|---|---|---|
| **Online training model** | | |
| Learning rate | 5e-5 | 2e-5 |
| Learning rate scheduler | Constant | Constant |
| LR warmup steps | 0 | 0 |
| Optimizer | AdamW | AdamW |
| AdamW - $\beta_1$ | 0.9 | 0.9 |
| AdamW - $\beta_2$ | 0.999 | 0.999 |
| Gradient clipping | 0.1 | 0.1 |
| **Discriminator** | | |
| Learning rate | 5e-5 | 5e-5 |
| Optimizer | AdamW | AdamW |
| AdamW - $\beta_1$ | 0 | 0 |
| AdamW - $\beta_2$ | 0.999 | 0.999 |
| Gradient clipping | 1.0 | 1.0 |
| Token loss weight $\alpha$ | 1e-3 | 1e-3 |
| Pixel loss weight $\beta$ | 5e-5 | 5e-5 |
| Adversarial loss weight $\lambda$ | 1 | 5e-1 |
| Gradient enable steps | 5 | 5 |
| Attribute concentration steps $r$ | 2 | 2 |
| LoRA rank | 128 | 128 |
| Classifier-free guidance scale | 7.5 | 7.5 |
| Resolution | $512 \times 512$ | $512 \times 512$ |
| Training steps | 2,000 | 2,000 |
| Local batch size | 4 | 6 |
| Local GT batch size | 2 | 2 |
| Mixed Precision | FP16 | FP16 |
| GPUs for Training | $8 \times$ NVIDIA A100 | $8 \times$ NVIDIA A100 |
| Training Time | $\sim$ 10 Hours | $\sim$ 24 Hours |

# F More Qualitative Results

## F.1 Effectiveness of the Fidelity Preservation module (FP) and Mixed Latent (ML) strategy

We visualize the effectiveness of how we preserve the generation capability of the diffusion model in Fig. 7. As shown in the figure, without any preservation technique, the diffusion model generates misshaped envelopes and swans. With the FP and ML applied, the diffusion model generates images aligned with the prompt and without artifacts.

## F.2 Comparison with the baseline model

We showcase more comparison results between our method with the baseline model in Fig. 11 to 14. Fig. 11 shows the generation results with long and complex prompts. Fig. 12 to 14 shows that our method solves various problems of misalignment, including object missing, incorrect attribute binding, incorrect relationship, inferior prompt understanding.

|                  SDXL                      CoMat-SDXL                    |                  SDXL                      CoMat-SDXL                    |

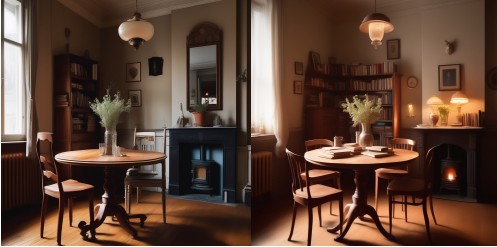 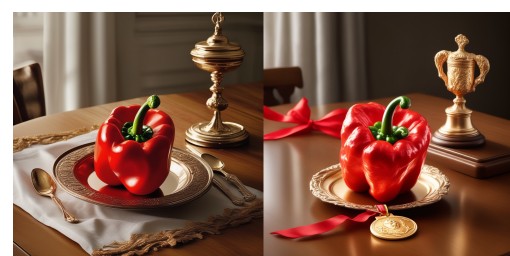

In the middle of a cozy room with a vintage charm, a circular wooden dining table takes the stage, its surface adorned with a decorative vase and **a few scattered books**. The room's warmth is maintained by an old-fashioned radiator humming steadily in the corner, a testament to its long service. **As dusk approaches, the waning sunlight softly permeates the space through a window** with a delicate frost pattern, casting a gentle glow that enhances the room's rustic ambiance.

A dining room setting showcasing an **unusually large** red bell pepper with a shiny, **slightly wrinkled texture**, prominently **placed beside a diminutive golden medal with a red ribbon** on a polished wooden dining table. The pepper's vibrant hue contrasts with the medal's gleaming surface. The scene is composed in natural light, highlighting the intricate details of the pepper's surface and the reflective quality of the medal.

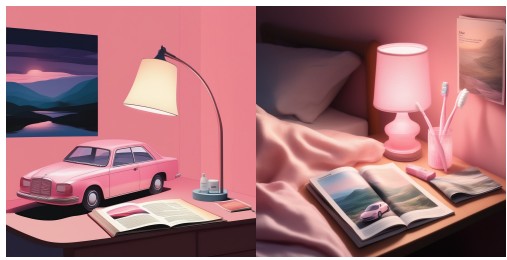 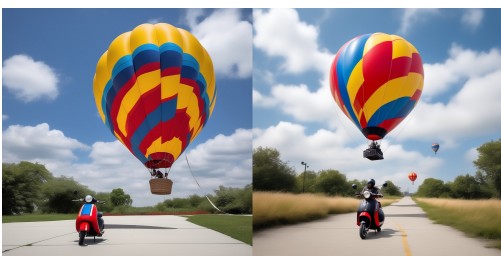

Inside a **dimly lit room**, the low luminance emanates from a bedside lamp casting a soft glow upon the nightstand. There lies a travel magazine, **its pages open to a vivid illustration of a car driving along a picturesque landscape**. Positioned next to the image is a **light pink toothbrush**, its bristles glistening in the ambient light. Beside the magazine, the **textured fabric of the bedspread is just discernible**, contributing to the composed and quiet scene.

A brightly colored hot air balloon with vibrant stripes of red, yellow, and blue hangs in the clear sky, its large round shape contrasting against the fluffy white clouds. Below it, a **sleek black scooter with red accents** speeds along a concrete pathway, **its rider leaning forward in a hurry**. The balloon moves at a leisurely pace, starkly contrasting with the frenetic energy of the scooter's rapid movement on the ground.

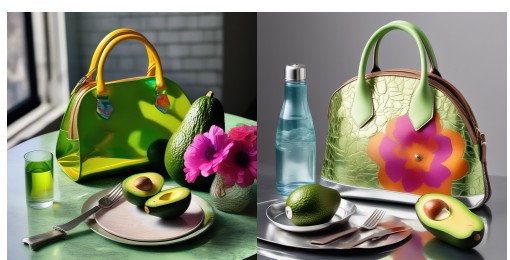 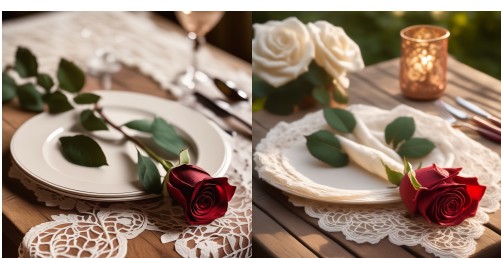

On a reflective **metallic table**, there is a **brightly colored handbag featuring a floral pattern** next to a freshly sliced avocado, its green flesh and brown pit providing a natural contrast to the industrial surface. The table is set for lunch, with **silverware** and **a clear glass water bottle** positioned neatly beside the avocado. The juxtaposition of the colorful fashion accessory and the rich texture of the avocado creates a striking visual amidst the midday meal setting.

A deep red rose with plush petals sits elegantly coiled atop an **ivory, intricately patterned lace napkin**. The napkin rests on a rustic wooden table that contributes to the charming garden setting. **As the late evening sun casts a warm golden hue over the area, the shadows of surrounding foliage dance gently around the rose**, enhancing the romantic ambiance. Nearby, **the green leaves of the garden plants** provide a fresh and verdant backdrop to the scene.

Figure 11: More Comparisons between SDXL and CoMat-SDXL on complex prompts. All pairs are generated with the same random seed.

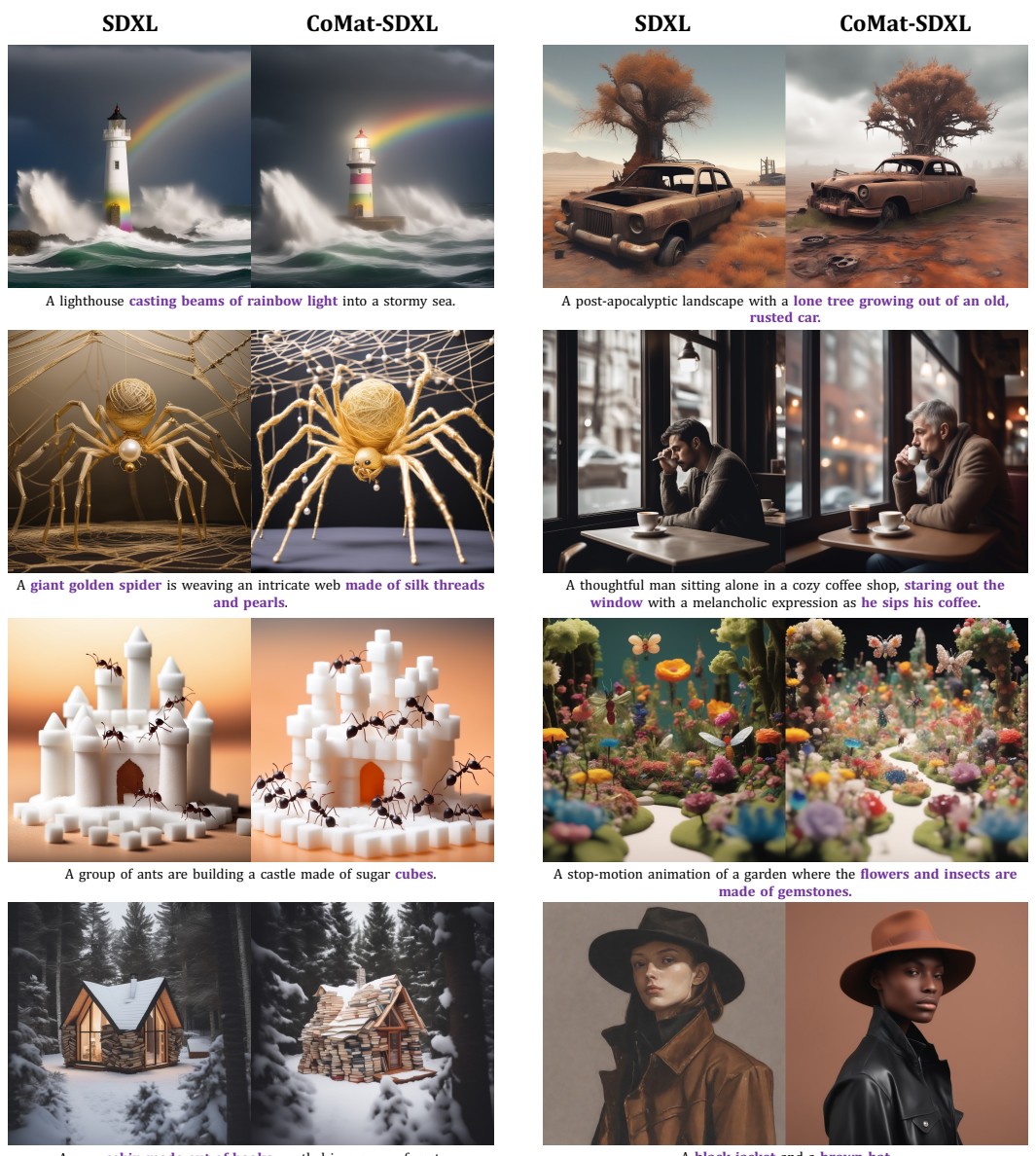

Figure 12: More Comparisons between SDXL and CoMat-SDXL. All pairs are generated with the same random seed.

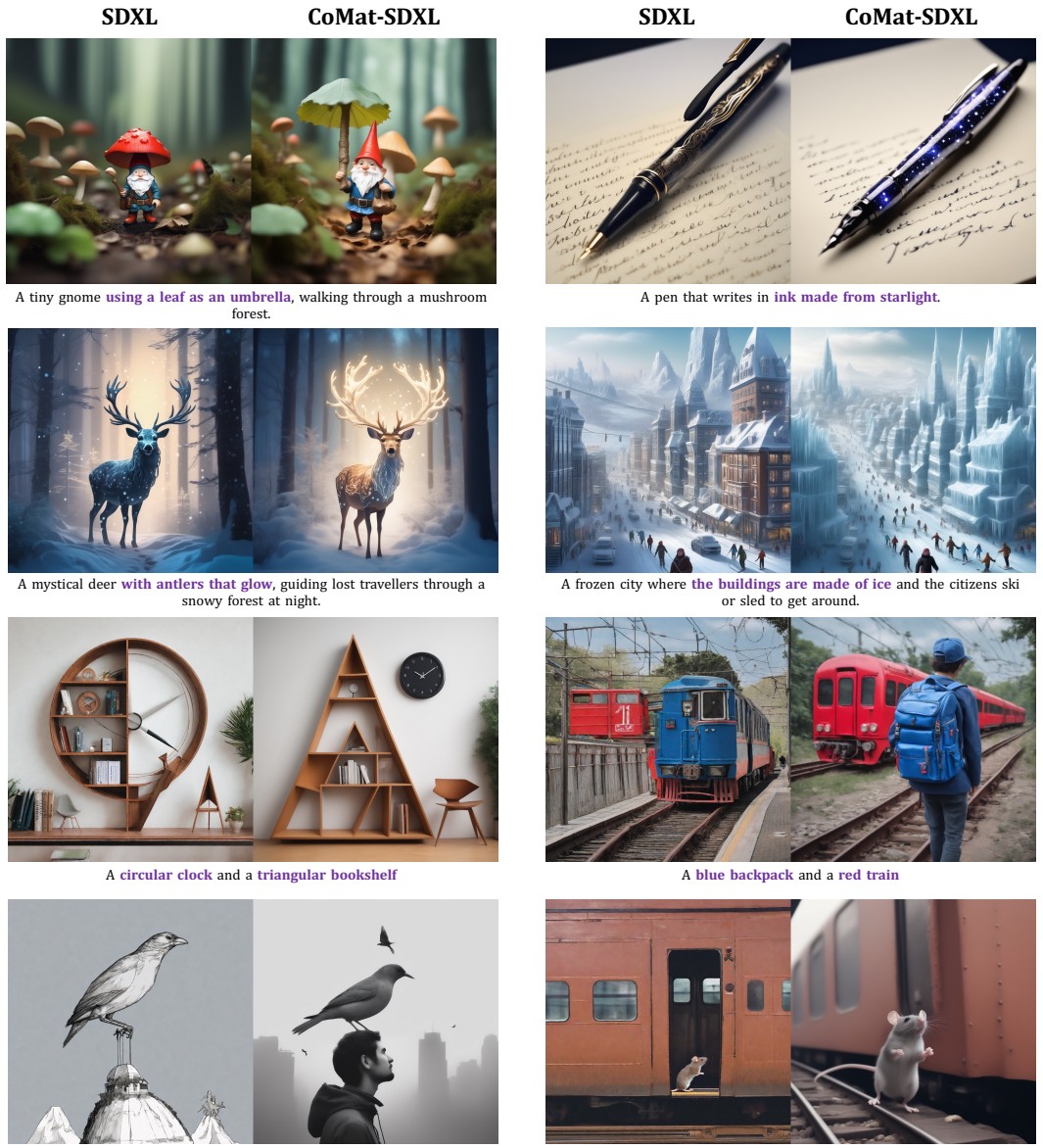

Figure 13: More Comparisons between SDXL and CoMat-SDXL. All pairs are generated with the same random seed.

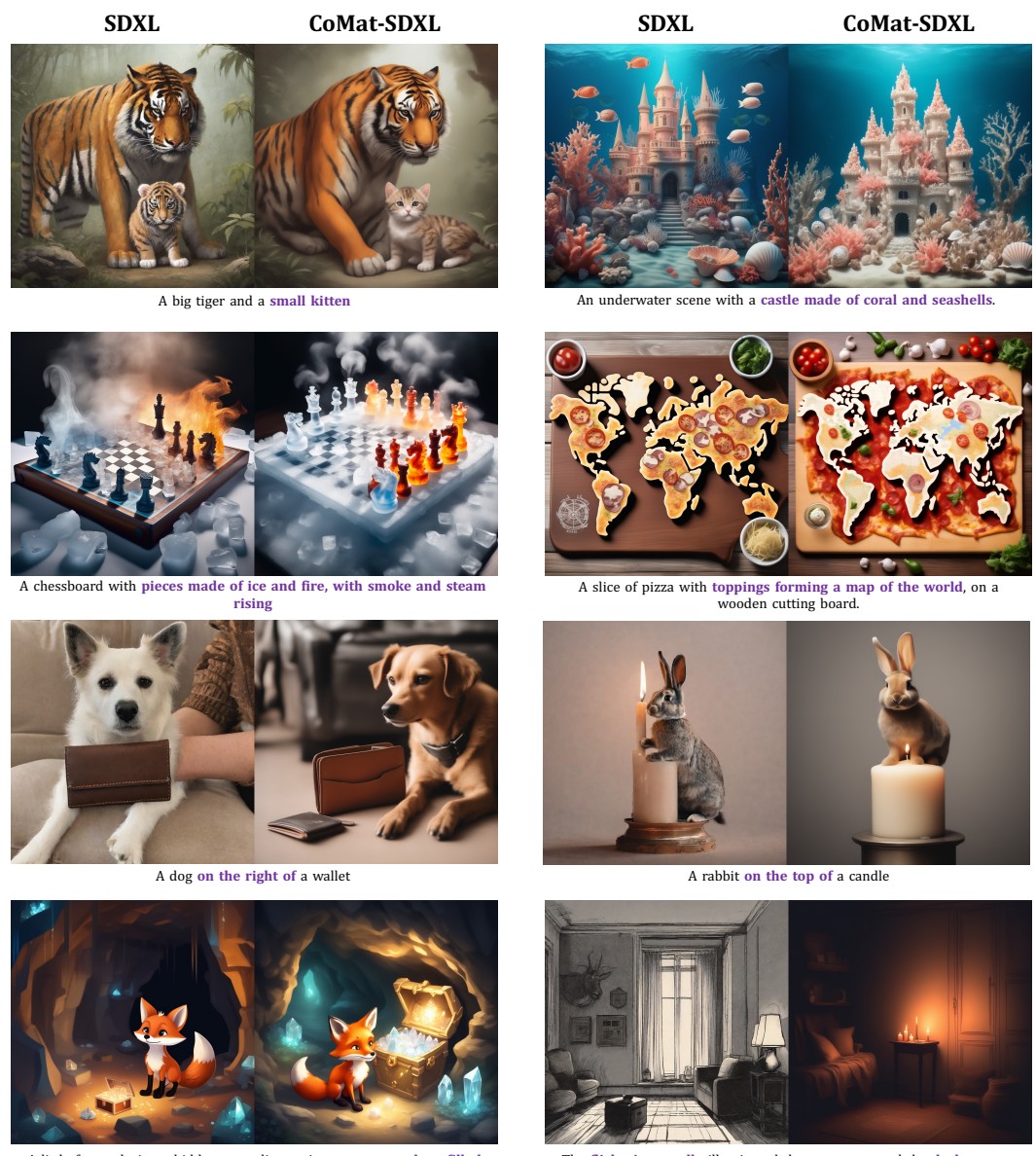

Figure 14: More Comparisons between SDXL and CoMat-SDXL. All pairs are generated with the same random seed.

