# OpenReview forum: "CoMat: Aligning Text-to-Image Diffusion Model with Image-to-Text Concept Matching"
_NeurIPS.cc/2024/Conference — NeurIPS 2024 poster_

### Official Review · Reviewer_9f7Y · 2024-07-10

**Soundness:** 2
**Presentation:** 3
**Contribution:** 3
**Rating:** 5
**Confidence:** 3

**Summary:**

The paper introduces CoMat, an end-to-end diffusion model fine-tuning strategy for text-to-image generation that addresses misalignments between text prompts and generated images. This method integrates a novel image-to-text concept activation module and an attribute concentration module, aimed at improving text-to-image alignment.

**Strengths:**

1. **Innovative Integration of Image-to-Text Models**: Utilizing an image-to-text model for concept activation in a diffusion-based approach is a novel application that enhances the alignment between generated images and text prompts.
2. **Comprehensive Modules for Fine-Tuning**: The attribute concentration module and the concept activation module are well-designed to tackle specific challenges in text-to-image generation, such as concept mismapping and concept omission.

**Weaknesses:**

1. **Complexity of Implementation**: The integration of multiple components such as image-to-text models, segmentation models, and the fine-tuning strategy may introduce significant complexity and computational overhead.
2. **Insufficient Comparative Analysis**: Although differences from methods like TokenCompose [1] are discussed, there is a notable lack of direct quantitative comparisons with state-of-the-art methods including TokenCompose [1], Structure Diffusion [2], and Attend-and-Excite [3]. This comparison is crucial to establish the relative performance and advancements over existing techniques.
3. **Resource Intensiveness of Loss Computation**: The loss computation method $L_{i2t}$ may be resource-intensive and inefficient as it requires the diffusion model to undergo multiple denoising steps to generate an image before the loss can be calculated, potentially impacting scalability and practical usage.

[1] Wang, Zirui, et al. "TokenCompose: Text-to-Image Diffusion with Token-level Supervision." Proceedings of the IEEE/CVF Conference on Computer Vision and Pattern Recognition. 2024.

[2] Feng, Weixi, et al. "Training-free structured diffusion guidance for compositional text-to-image synthesis." arXiv preprint arXiv:2212.05032 (2022).

[3] Chefer, Hila, et al. "Attend-and-excite: Attention-based semantic guidance for text-to-image diffusion models." ACM Transactions on Graphics (TOG) 42.4 (2023): 1-10.

**Questions:**

Can the image-to-text model **effectively** offer pixel-level optimization guidance to the diffusion model using $L_{i2t}$? Additionally, are there any ablation studies that demonstrate the effects of using $L_{i2t}$ for training?

**Limitations:**

The proposed CoMat model incorporates several sophisticated components, including an image-to-text concept activation module and an attribute concentration module. While these additions are innovative, they significantly increase the model's complexity and computational demands. This complexity may limit the scalability of the approach, especially in resource-constrained environments or when handling large-scale datasets.

---

> ### Author Rebuttal · Authors · 2024-08-07
>
> We sincerely appreciate your valuable comments. We found them extremely helpful in improving our draft. We address each comment in detail, one by one below.
>
> **Comment 1.  Complexity of Implementation**
>
> Thank for your feedback. Indeed, we acknowledge the intricate nature of our approach. However, we argue that the complex design is necessary. Our method's multiple components work as a whole to  address the misalignment problem while maintaining high-quality image generation:
>
> 1. Misalignment Solution: a) Concept Activation Module: Utilizes an image-to-text model to activate individual concepts within the prompt. b) Attribute Concentration: Employs a segmentation model to guide the activated concepts to their appropriate spatial locations in the generated image.
> 2. Quality Preservation: a) Fidelity Preservation Module: Incorporates a pre-trained diffusion model as a discriminator to mitigate potential quality degradation resulting from the image-to-text model's guidance. b) Mixed Latent Strategy: Leverages ground truth images to provide additional guidance during the diffusion process.
>
> As for the resource overhead, instead of full finetuning, our method uses LoRA to reduce the computational cost. Please refer to **the response to Comment 3** for further discussion.
>
> **Comment 2. Insufficient Comparative Analysis**
>
> Thanks for your advice. We provide more quantitative comparisons with state-of-art methods like TokenCompose [1], Structure Diffusion [2], and Attend-and-Excite [3] in the T2I-CompBench benchmark, as shown in Table 1 in the author rebuttal PDF. We will include these results in our final draft.
>
> **Comment 3. Resource Intensiveness of Loss Computation**
>
> Sorry for the confusion caused. We respectfully disagree for the following two reasons:
>
> 1. **The denoising process is a must for all the training losses, not only $\mathcal{L}_{i2t}$.**
>
>    Our training process indeed needs multiple denoising steps to generate the image. However, we add multiple supervision during this process to foster the model to align the generated image with the prompt.
>
>    The multiple denoising process starts from a pure noise $x_T$, and then we iteratively denoise the noise to produce an image $\mathcal{I}$. Of all the denoising steps, we uniformly sample $K$ steps to enable the gradient.  We supervise the attention maps during denoising in these steps, which corresponds to $\mathcal{L}\_\text{pos}$  and   $\mathcal{L}\_\text{neg}$. The gradient of  $\mathcal{L}\_{i2t}$ and  $\mathcal{L}\_{adv}$ is back-propagated through the image to the LoRA for the sampled $K$ steps.
>
> 2. **More efficient and beneficial proven in previous works**
>
>    Compared with directly supervised fine-tuning the diffusion model, our training process is identical to the inference process of the diffusion model. This training and test alignment contributes to a more efficient learning process. In fact, our method only needs 2000 iters to achieve a good performance, while directly finetuning typically requires much more iteration (Please refer to the table in Limitation 1). This paradigm of training has been proven more efficient and beneficial in previous works like DPOK [5].
>
> **Question 1. Can image-to-text model offer effective pixel-level guidance?**
>
> We provide the visualization of the gradient of $\mathcal{L}\_{i2t}$ on the generated image in the Fig. 1 in the author rebuttal PDF.
>
> **Question 2. Ablation studies that demonstrate the effects of using $\mathcal{L}_{i2t}$ for training**
>
> We conduct ablation studies on the T2I-CompBench to verify the effectiveness of the $\mathcal{L}\_{i2t}$ losses in Table 4 in the paper. The $\mathcal{L}\_{i2t}$ greatly enhances the text-image alignment of the diffusion model. Please refer to Section 5.4  for more details.
>
> **Limitation 1. Increase model's complexity and computational demands**
>
> Thanks for your feedback. Although our method does introduce multiple components, these components bring about abundant supervision to foster the fast training speed and also excellent performance. Besides, we list the performance on T2I-CompBench and training cost compared with GORS [4]:
>
> | Model       | Iteration    | GPU Num | Color$\uparrow$ | Shape$\uparrow$ | Texture$\uparrow$ | Spatial$\uparrow$ | Non-Spatial$\uparrow$ | Complex$\uparrow$ |
> | ----------- | ------------ | ------- | --------------- | --------------- | ----------------- | ----------------- | --------------------- | ----------------- |
> | GORS (SDv2) | 50000-100000 | 8       | 0.6603          | 0.4785          | 0.6287            | 0.1815            | 0.3193                | 0.3328            |
> | Comat-SD1.5 | 2000         | 8       | **0.6734**      | **0.5064**      | 0.6243            | **0.2073**        | 0.3166                | **0.3575**        |
>
> Our method well balances the training iterations and the performance. Comat-SD1.5 generally achieves better performance with roughly 2% of iterations.
>
> Besides, all these modules are removed during inference. Therefore, no inference cost is introduced.
>
> [1] Wang, Zirui, et al. "TokenCompose: Text-to-Image Diffusion with Token-level Supervision." Proceedings of the IEEE/CVF Conference on Computer Vision and Pattern Recognition. 2024.
>
> [2] Feng, Weixi, et al. "Training-free structured diffusion guidance for compositional text-to-image synthesis." arXiv preprint arXiv:2212.05032 (2022).
>
> [3] Chefer, Hila, et al. "Attend-and-excite: Attention-based semantic guidance for text-to-image diffusion models." ACM Transactions on Graphics (TOG) 42.4 (2023): 1-10.
>
> \[4\] Huang, Kaiyi, et al. "T2i-compbench: A comprehensive benchmark for open-world compositional text-to-image generation." *Advances in Neural Information Processing Systems* 36 (2023): 78723-78747.
>
> \[5\] Fan, Ying, et al. "Reinforcement learning for fine-tuning text-to-image diffusion models." *Advances in Neural Information Processing Systems* 36 (2024).

---

> > ### Comment · Reviewer_9f7Y · 2024-08-13
> >
> > Thank you for your response. Most of my concerns have been addressed. However, I'm still unsure whether the image-to-text model provides effective optimization guidance as presented in Fig. 1 of the author rebuttal PDF. Additionally, I have some concerns about the complexity of the method design. Despite these reservations, I believe this is solid work. Therefore, I’ve decided to raise my score to borderline accept in recognition of your efforts. Thank you for your hard work.

---

> > > ### Author Response · Authors · 2024-08-13
> > >
> > > We sincerely thank the reviewer for the kind support of our work!
> > >
> > > **A reasonable explanation to effective optimization guidance of the image-to-text model**
> > >
> > > The image-to-text model is trained to predict the caption, which is highly aligned with the given image. Therefore, once the generated image is not aligned with the prompt, the image-to-text model will be "reluctant" to output the prompt as the caption for the generated image. This "reluctance" comes from the misalignment of certain concepts in the image. The image-to-text model views the image, discovers the misalignment areas, and finally gives a high $\mathcal{L}\_\{i2t}$. We leverage this attribute and try to minimize this "reluctance". The gradient of  $\mathcal{L}\_\{i2t}$ mainly summits at the areas where misalignment occurs, as shown in Fig.1 in the author rebuttal PDF. This gradient is further back-propagated to the diffusion model to fix the misalignment of this area. That's where the pixel-level guidance comes from.
> > >
> > > We will definitely make this point clearer in the revised version!

---

### Official Review · Reviewer_3go2 · 2024-07-11

**Soundness:** 2
**Presentation:** 3
**Contribution:** 3
**Rating:** 6
**Confidence:** 3

**Summary:**

The paper breaks down the misalignment problem in T2I two: concept ignorance and concept mismapping, and propose a fine-tunning strategy to enhance the prompt understanding and following.
The methods include two modules: The concept activation module to maximize the posterior probability; An attribute concentration module is proposed for positive and negative mapping.

**Strengths:**

This paper proposes a general framework for supervising image generation, including a Image to text model scoring method and a prior preservation loss for better text alignment without losing fidelity; and a attribute concentration method which utilizes open vocabulary segmentation method to force attention modification area.

**Weaknesses:**

1. The training cost is disproportionate to this complicated framework. This framework includes diffusion training - full steps inference - mllm judging - grounding segmentation for each entity ... However, the training cost is only ten hours for 8A100 and only needs 2k iterations to converge, and 128 rank lora is enough. I doubt the limited training is enough to convert the SDXL model to attain the attribute learning ability therefore the soundness of this paper. It is needed to provide more training information to testify the experiments are enough and real.

2. The real training cost is a concern since the training involves manipulating diffusion results in the pixel space and inference with MLLM.

3. The proposed method seems cannot assign attributes to multiple same-name objects, such as an Asian girl with an Indian girl, the segmentation model cannot differentiate two girls and therefore cannot assign attributes.

Others are questions not weakness, please see the questions part

**Questions:**

1. how to use image-to-text LLVM to provide a matching score?, since most LLVM models are developed for VQA and captioning, scoring is usually not a skill for them.

2. The description of Mixed Latent Strategy is not sufficient. After reading twice, I didn't get the meaning of  "in addition to the latent starting from pure noise", T2i training never starts from pure noise but from $x_0 + \epsilon_{t}$,

3. Since the training details concern me, if authors can kindly open-source the code would make the paper's soundness no concern. Would you open-source the code?

**Limitations:**

Yes, to a certain extent.. The paper doesn't contain a limitation section in the main paper. However, in supplementary material, training cost and mllm usage concern is discussed.

---

> ### Author Rebuttal · Authors · 2024-08-07
>
> We sincerely appreciate your valuable comments. We found them extremely helpful in improving our draft. We address each comment in detail, one by one below.
>
> **Comment 1. Training cost is disproportionate**
>
> Thank you for your feedback. We will open-source our training code for reproducibility.
>
> We test on more training iterations in our pilot study but no apparent gain is witnessed. So we keep 2000 iterations as default. Also, we test the different settings for training parameters: rank 128, 256 for LORA, and full parameter finetuning. We find no apparent gain for larger rank or full parameter fintuning. We finally choose LoRA with rank 128 for efficient training. LoRA of relatively small rank could already convert the model with different abilities. Various previous works adopt this setting and achieve good performance [1, 2].
>
> **Comment 2. Real training cost**
>
> Thank you for your feedback.
>
> 1. Pixel space supervision
>
>    During the full steps inference, we uniformly sample $K$ steps to perform the pixel space supervision, namely calculating the $\mathcal{L}\_\text{pos}$ and  $\mathcal{L}\_\text{neg}$ loss. The $K$ is 5 in our experiments. Therefore, only 10% of steps in the inference require gradient and pixel space supervision. This may partially account for the fast training speed.
>
> 2. MLLM inference
>
>    In practice, we leverage BLIP trained on captioning tasks instead of MLLM (Please refer to Appendix A.3 in the main paper for detailed information). This choice accelerates the computation of the $\mathcal{L}\_{i2t}$ loss.
>
> **Comment 3. Assign attributes to multiple same-name objects**
>
> This is a really interesting question. Currently, it is difficult to assign the attribute to multiple same-name objects by only using the segmentation model. However, we argue that the image-to-text model with advanced image-text understanding ability may distinguish the attribute for each same-name object and offer valid guidance. As shown in the Fig. 1 in the author rebuttal PDF, the image-to-text model could also provide pixel-level supervision to the image.
>
> We will work on to solve this question in our future work.
>
> **Question 1. How to use image-to-text LLVM to provide a matching score?**
>
> Sorry for the confusion caused. The scoring is conducted by calculating the loglikelihood for the LLVM to output the prompt as the caption, given the generated image.
>
> In practice, we leverage the LLVM specifically trained for the captioning tasks (e.g., BLIP) to do the scoring. Their scoring capability comes with their training nature. These captioning models are trained with the negative loglikelihood loss, namely, the model needs to maximize the probability for generating the caption given the corresponding image. Therefore, whenever the generated image does not align with the text prompt, the LLVM will output low loglikelihood, namely, the $\mathcal{L}\_{i2t}$ is big. We treat the $-\mathcal{L}\_{i2t}$ as the alignment score of LLVM and try to maximize it, i.e., minimize $\mathcal{L}\_{i2t}$.
>
> **Question 2. More details about Mixed Latent Strategy**
>
> Sorry for the confusion caused. We detail our mixed latent strategy below:
>
> The mixed latent strategy contains two types of latents in the fine-tuning procedure, i.e., the latent starting from the pure noise and the noisy latent from the GT Images.
>
> 1. **Latent starting from the pure noise**
>
>    This serves as the main branch in our pipeline.
>
>    Our fine-tuning process shares the same procedure to generate an image as the diffusion model does in the inference time. We uniformly sample $K$ steps from all the inference steps to enable the gradient. Therefore, the latent is sampled from the pure noise $\mathcal{N}(0,I)$. We iteratively denoise it to obtain the generated image. The image is then used to calculate the $\mathcal{L}\_{i2t}$ and $\mathcal{L}\_{adv}$ loss. It is also sent to the segmentation model to provide the object mask for computing the $\mathcal{L}\_\text{pos}$  and $\mathcal{L}\_\text{neg}$.
>
>    The latent starting from the pure noise corresponds to the upper left part in Fig. 4. in the main paper.
>
>    Please refer to [2] for how to receive the gradient from the loss.
>
> 2. **Noisy latent from the GT Images**
>
>    We also aim to inject information from the GT images to stabilize the fine-tuning process.
>
>    We randomly sample a timestamp $\tau$ from a pre-defined range $[T\_1, T\_2]$. Then we obtain $x\_{\tau}$ by adding the timestamped noise $\epsilon\_{\tau}$ on the latent of the GT Image $x_0$. We also iterativaly denoise this noisy GT latent to get $\hat{x}\_0$ as we do for the latents starting from the pure noise. This $\hat{x}\_0$ is only used to calculate the $\mathcal{L}\_{i2t}$ loss.
>
>    The latent starting from the noisy GT corresponds to the upper left part in Fig. 4 in the main paper.
>
> **Question 3. Will the code be open-sourced?**
>
> Thanks for your feedback. We will open-source the training code.
>
>
>
> [1] Sun, Jiao, et al. "Dreamsync: Aligning text-to-image generation with image understanding feedback." *Synthetic Data for Computer Vision Workshop@ CVPR 2024*. 2023.
>
> [2] Wu, Xiaoshi, et al. "Deep Reward Supervisions for Tuning Text-to-Image Diffusion Models." *arXiv preprint arXiv:2405.00760* (2024).

---

> > ### Comment · Reviewer_3go2 · 2024-08-13
> > **Thanks for your detailed reply, i'll keep my score as Borderline accept**
> >
> > This paper needs a lot of effort to build the whole pipeline and the result's soundness is credible which makes me maintain the accepted score.
> >
> > On the other hand, the paper's innovation is more about the engineering part with code not open-sourced. So I decided to maintain my score as Borderline accept

---

> > > ### Author Response · Authors · 2024-08-13
> > >
> > > We sincerely thank the reviewer for their support and valuable feedback on our work!
> > >
> > > We appreciate your suggestions, particularly regarding the clarification of LVLM scoring usage and the mixed latent strategy. These insights will significantly strengthen our paper, and we will incorporate detailed explanations in our revision. Additionally, we will work on to address the issue of assigning attributes to multiple objects with the same name. We believe these revisions above will substantially enhance the quality and contribution of our paper.
> > >
> > > We are committed to open-sourcing all of our training codes to ensure full reproducibility of the training process upon acceptance. Given this commitment and our planned revisions, we respectfully ask if you would consider reevaluating our work and raising the score.

---

> > > > ### Comment · Reviewer_3go2 · 2024-08-13
> > > >
> > > > It's hard for me to raise my score to Weak Accept (although I did at last). CoMat is a complicated framework which needs a lot of engineering techniques if this is an open source project, I think it would be great for the community.

---

> > > > > ### Author Response · Authors · 2024-08-13
> > > > >
> > > > > Thank you for your valuable support of our research!
> > > > >
> > > > > We commit to open-sourcing all training codes upon acceptance of our work.

---

### Official Review · Reviewer_82oN · 2024-07-11

**Soundness:** 4
**Presentation:** 4
**Contribution:** 3
**Rating:** 5
**Confidence:** 5

**Summary:**

This paper proposes a fine-tuning strategy for text-to-image diffusion models to improve the alignment of generated images to text prompts. The solution components are summarized as follows:
1. Concept Activation Module: This module helps the model focus on ignored text concepts by leveraging an image-to-text model to supervise the generation process
2. Attribute Concentration Module: This module aims to improve the localization of attributes within images, ensuring that characteristics like color or texture are correctly applied to the right parts of the image.

**Strengths:**

1. An end-to-end fine-tuning strategy is employed to address various text-image misalignment issues in a unified manner, making it easy to deploy.

**Weaknesses:**

1. Functionally, the Fidelity Preservation proposed in this paper is similar to the Class-specific Prior Preservation Loss introduced in reference [1]. Please provide a detailed analysis of the differences between the two.

2. The model presents a potential risk of overfitting the characteristics of the training data, which may affect its generalizability to various real-world scenarios. Therefore, additional experiments related to generalization are necessary. For instance, it should be evaluated whether the model fine-tuned on dataset A can demonstrate improvements when tested on dataset B.

3. The misalignment between generated images and textual concepts is a complex issue with potentially intricate underlying causes. Therefore, it is recommended to use the methods proposed in reference [2]  to identify any remaining misalignment issues in the fine-tuned model.


[1] Ruiz, Nataniel, et al. "Dreambooth: Fine tuning text-to-image diffusion models for subject-driven generation." Proceedings of the IEEE/CVF conference on computer vision and pattern recognition. 2023. \
[2] Du, Chengbin, et al. "Stable diffusion is unstable." Advances in Neural Information Processing Systems 36 (2024).

**Questions:**

See weakness.

**Limitations:**

Already discussed in the paper

---

> ### Author Rebuttal · Authors · 2024-08-07
>
> We sincerely appreciate your valuable comments. We found them extremely helpful in improving our draft. We address each comment in detail, one by one below.
>
> **Comment 1.  Differences between Class-specific Prior Preservation Loss with Fidelity Preservation**
>
> Thank you for your advice. Indeed, our Fidelity Preservation (FP) module shares a similar high-level idea with the Class-specific Prior Preservation Loss (CPP Loss) introduced in [1], i.e., preserving the generation quality while finetuning. However, our method is very different in the following two aspects:
>
> 1. **Target Task and Preserve Domain**
>
>    DreamBooth seeks to personalize image generation for specific objects. While the introduced CPP Loss primarily maintains generative capabilities within a narrow domain—specifically, the object class present in the training data—our proposed FP module operates within the context of text-image alignment. FP aims to preserve general generative capabilities by computing adversarial loss across the entire training dataset, encompassing a diverse range of text prompts.
>
> 2. **Methodology**
>
>    DreamBooth finetunes the diffusion model with the pretraining loss, i.e., the squared error denoising loss on a certain timestamp. CPP Loss follows its form.
>
>    In contrast, our fine-tuning procedure simulates the inference process of the diffusion model to conduct a full-step inference. We aim to directly supervise the generated image to achieve the training-test alignment. Therefore, we propose the novel FP module to leverage a discriminator to adversarially preserve its quality. The applied discriminator is also updated along with the fine-tuning process, enabling finer control of the image quality.
>
> We will add this discussion to the final draft for clarification.
>
> **Comment 2. Potential risk of overfitting the training data**
>
> Thank you for your feedback. We conduct zero-shot evaluation on long and complex prompt in DPG-Bench in the Table 2 in our main paper.
>
> All of our training data contains only short text prompts similar to COCO captions. The length of sentences in the training set is 12.13 words on average. The average prompt length in DPG-Bench is 78.23 words.
>
> As shown in the Table 2 in the main paper, our method significantly enhances the alignment by over 10 points on SD1.5 and over 2 points in SDXL. The result proves that our proposed method improves the general prompt-following capability instead of overfitting to the training data. The qualitative example on DPG-Bench is shown in Fig. 11 in the Appendix.
>
> Besides, TIFA in Table 2 in the main paper is another alignment benchmark, which our training data also does not overlap. We witnessed a 7.4 points improvement in Comat-SD1.5 and a 1.6 points improvement in Comat-SDXL. The result further reveals the generalization.
>
> **Comment 3. Identify remaining misalignment issues in the fine-tuned model**
>
> Thank you for your advice. This is a very interesting suggestion. We investigate our method proposed in [2]. We will add an extra section in the appendix to discuss our findings.
>
>
>
> [1] Ruiz, Nataniel, et al. "Dreambooth: Fine tuning text-to-image diffusion models for subject-driven generation." Proceedings of the IEEE/CVF conference on computer vision and pattern recognition. 2023.
>
> [2] Du, Chengbin, et al. "Stable diffusion is unstable." Advances in Neural Information Processing Systems 36 (2024).

---

> > ### Comment · Reviewer_82oN · 2024-08-13
> >
> > Since my concerns have not been fully addressed, I have decided to keep the score unchanged. For example, the author did not provide a positive response to Q3, not even a brief experimental analysis.

---

> > > ### Author Response · Authors · 2024-08-14
> > >
> > > Thank you for your feedback. We sincerely apologize for the concise response and any misunderstanding it may have caused.
> > >
> > > **We initiated testing of our method using the referenced work [1] at the start of the rebuttal period. However, due to time constraints and limited GPU resources, we were unable to complete the experiments before the rebuttal phase concluded.** Consequently, we plan to present the full extent of our findings in the final draft.
> > >
> > > Below, we provide details of the experiments conducted and **our current observations** to offer insights into the remaining misalignments we've identified with the assistance of [1].
> > >
> > > First of all, we briefly summarize [1]: the paper introduces an attack method to automatically find out what prompt caused Stable Diffusion model to generate misalignment images. Based on the successful attack prompts, [1] summarizes four prompt patterns where Stable Diffusion model often fails to generate aligned images. The four patterns are: a) Variability in Generation Speed; b) Similarity of Coarse-grained Characteristics; c) Polysemy of Words; d) Positioning of Words.
> > >
> > > Our experiment and analysis contains the following 3 parts:
> > >
> > > **Experiment 1. Do the discovered four patterns of the original impairment of Stable Diffusion model still exist in CoMat?**
> > >
> > > Based on the example four patterns in [1], we manually create 20 prompts for the patten (a), (b) and (c), and 5 meta patterns for pattern (d), which generates 30 prompts. Then we use them as prompts for the original Stable Diffusion model and CoMat. The generated images are evaluated by humans. We show the ratio of successful generation below:
> > >
> > > | Model                 | Pattern (a) | Pattern (b) | Pattern (c) | Pattern (d) |
> > > | --------------------- | ----------- | ----------- | ----------- | ----------- |
> > > | Stable Diffusion v1.5 | 45.0%       | 35.0%       | -           | 13.3%       |
> > > | CoMat-SD1.5           | 65.0%       | 50.0%       | -           | 30.0%       |
> > >
> > > This result align with the enhancements targeted by CoMat. Pattern (a), (b) and (d) all involves multiple objects in the prompt. The Concept Activation Module contributes to the object's existence. Besides, the Attribute Concentration Module restricts the diffusion model to only attend to one object's token for each object area in the image. This further prevents the object combining problem found in pattern (b).
> > >
> > > The pattern (c) is about the polysemy of words. We argue that without sufficient contextual information, it is infeasible to evaluate the generation result.
> > >
> > > This experiment demonstrates that CoMat greatly mitigates the original vulnerabilities discovered in Stable Diffusion. However, we acknowledge that there remains a gap between CoMat's performance and perfect alignment, particularly in addressing pattern (c) where prompts contain more than two objects. We are actively working to address these remaining challenges. We apologize that the visualization result cannot be provided in the discussion stage, we will include the visualization result in our final draft.
> > >
> > > **Experiment 2. Can the learned Gumbel Softmax distribution of Stable Diffusion v1.5 still effectively attack CoMat-SD1.5?**
> > >
> > > We implement the method of [1] using its open-sourced code in GitHub. Following the setting in [1], we learn a Gumbel Softmax distribution for each class from ImageNet-1K [2]. Due to the time limit, instead of generating 50 images for each class, we generate 5 images for each class and calculate the attack success rate. Then, we directly use this learned Gumble Softmax distribution to attack CoMat-SD1.5. The result is below:
> > >
> > > | Model                 | Short Prompt Success | Long Prompt Success |
> > > | --------------------- | -------------------- | ------------------- |
> > > | Stable Diffusion v1.5 | 47.6%                | 51.1%               |
> > > | CoMat-SD1.5           | 39.8%                | 45.3%               |
> > >
> > > As the result shown, CoMat-SD1.5 suffers less from the attack. This result is reasonable since CoMat has gone over the fine-tuning process so the distribution for SD1.5 may be not valid for CoMat-SD1.5.
> > >
> > > We will include the result following the original setting in [1], where 50 samples are generated for each class, in our final draft.
> > >
> > > **Experiment 3. How does CoMat act under the auto-attack method proposed in [1]?**
> > >
> > > Finally, we directly apply the method introduced in [1] to CoMat-SD1.5. Again, due to the time and resource limit, we only test on 5 samples for each class. The result is below:
> > >
> > > | Model       | Short Prompt Success | Long Prompt Success |
> > > | ----------- | -------------------- | ------------------- |
> > > | CoMat-SD1.5 | 45.2%                | 52.3%               |
> > >
> > > Our experiments reveal that CoMat demonstrates superior robustness compared to SD1.5 in scenarios involving short prompt attacks. However, we observed a decline in performance when dealing with long prompts. This discrepancy may be due to the lack of long prompts in CoMat's training dataset.

---

> > > > ### Author Response · Authors · 2024-08-14
> > > >
> > > > Overall, we sincerely apologize for not timely providing the final result of analysis and experiment due to the time and resource limitations. However, we believe the current result delivers insights of the misalignment still present in CoMat. We will cite [1] in the experiment part to discuss these insights in our main paper in the final draft. Hopefully, this can address your concern. We respectfully ask if you would consider reevaluating our work and raising the score.
> > > >
> > > > [1] Du, Chengbin, et al. "Stable diffusion is unstable." Advances in Neural Information Processing Systems 36 (2024).
> > > >
> > > > [2] Russakovsky, Olga, et al. "Imagenet large scale visual recognition challenge." *International journal of computer vision* 115 (2015): 211-252.

---

> > > > ### Comment · Reviewer_82oN · 2024-08-14
> > > >
> > > > In Experiment 3, it was shown that after the model was fine-tuned with CoMat, the attack success rate of long prompts actually increased. The authors believe that this discrepancy may be due to the lack of long prompts in CoMat's training dataset. Coupled with the second issue I initially raised, namely the potential risk of overfitting the training data, could this suggest that the method has a weakness in generalization to some extent?

---

> ### Author Response · Authors · 2024-08-14
>
> Thanks for your response.
>
> We also conduct zero-shot evaluations on the DPG-Bench, which consists of long and complex prompts. Our method exhibits excellent zero-shot performance both qualitatively and quantitatively. Please refer to our rebuttal for Comment 2 for details. We believe this could address the concern of the generalization ability to the long prompts. Besides, we visualize some successful attacks, and we find that the generated prompt contains many meaningless letters, which could largely affect the generation process. An example is: "these small, colorful fiddler crab are known for their distinctively asymmetric claws, with one being much larger than the other. males use their enlarged claw to attract mates and defend their territory., hair, joheat col< troll < shepherds a < ; children "" < lake rest <". Therefore, we argue that the result in Experiment 3 may not fully represent the result of our method's generalization ability.

---

> ### Comment · Reviewer_82oN · 2024-08-14
>
> There is some controversy regarding the issue of generalization, but this is not a critical flaw. The experimental results with short prompts do show that the model fine-tuning has improved text-image alignment to some extent (although it is difficult to determine the extent of this improvement given the limited amount of data). Therefore, I have decided to raise my score to a Borderline accept.

---

### Official Review · Reviewer_KjKz · 2024-07-14

**Soundness:** 3
**Presentation:** 3
**Contribution:** 2
**Rating:** 4
**Confidence:** 4

**Summary:**

This paper studies the prompt-following issues within text-to-image generation models and proposes a very simple yet effective solution by supervising the diffusion models with recognition models like BLIP for image captioning and Grounded-SAM for image segmentation. The authors also propose a fidelity preservation strategy and a mixed latent strategy to address overfitting risks. They fine-tune the SDXL or SD1.5 models on 20K complex text prompts and demonstrate encouraging results on various benchmarks that focus on evaluating attribute binding and object relationship accuracy.

**Strengths:**

- The idea of fine-tuning text-to-image generation models with image recognition models, which act as a reward model, is reasonable and has been explored in previous works such as AlignProp [1] and DRaFT [2].

[1] Aligning Text-to-Image Diffusion Models with Reward Backpropagation, arXiv 2023

[2] Directly Fine-Tuning Diffusion Models on Differentiable Rewards, ICLR 2024

- The experimental results effectively verify the proposed method's effectiveness.

**Weaknesses:**

- The technical novelty is limited, as similar ideas have been explored in both AlignProp [1] and DRaFT [2]. The key difference lies mainly in the different choices of text prompts and the additional supervision on the attention maps.
- The authors fail to provide a thorough user study to justify the effectiveness of the proposed approach from the user's perspective.
- Although the proposed approach achieves encouraging results, these results focus primarily on fine-grained text-following capability rather than more fundamental challenges such as counting-aware prompts and spatial-aware arrangement prompts. Consequently, the actual contribution of this paper is relatively weak, as the presented results focus on improving benchmark performance rather than addressing the core challenges within text-to-image generation models.

**Questions:**

Refer the Weaknesses

**Limitations:**

Refer the Weaknesses

---

> ### Author Rebuttal · Authors · 2024-08-07
>
> We sincerely appreciate your valuable comments. We found them extremely helpful in improving our draft. We address each comment in detail, one by one below.
>
> **Comment 1. Limited technical novelty**
>
> Thank you for your feedback. We respectfully disagree with the reviewer's assessment. Indeed, our method follows the basic reward fine-tuning pipeline explored in [1, 2]. **We argue that we only adopt this basic paradigm for fine-tuning the diffusion model. We introduce a variety of novel designs in our procedure to specifically solve the underlying problems in the image-text alignment issue. It is inappropriate to simply classify our method as the same approach of TODO(AlignProp, Draft).** As stated in the strength section of reviewer 9f7Y, it is innovative to introduce the image-to-text model to solve the image-text alignment task. And the other modules are well-designed for tackling specific challenges in image-text alignment task.
>
> We summarize our technical novelty as follows:
>
> 1. **Image-to-text model as reward model**
>
>    We investigate how to use the hidden knowledge of the pre-trained image-to-text model to supervise the diffusion model for alignment. We extend the basic reward fine-tuning pipeline to the image-text alignment task and prove that the pre-trained image-to-text model is a suitable reward model for a large setting of training.
>
> 2. **Attribute Concentration module**
>
>    Our visualisation results reveal the activated concepts often fails to mapping to the correct area in the image, which causes the misalignment. We introduce the novel attribute concentration module to supervise on the pixel level to address this issue.
>
>    Previous works like [1, 2] only supervises the generated image, omitting guiding the middle steps when generating. How to add supervision for finer-grained control in these steps has not been explored before our method.
>
> 3. **Fidelity Preservation module**
>
>    [1, 2] either focus on the aesthetic task or is limited to a very small setting (tens of prompts for the training data). How to effectively preserve the generation capability in a larger setting remains unexplored. We introduce the novel Fidelity Preservation module to leverage the underlying knowledge of the pre-trained diffusion model to be the discriminator. This adversarial design effectively solves the image corruption.
>
> 4. **Mixed Latent Strategy**
>
>    We also novelly propose to mix the noisy GT latent with the latents starting from pure noise. This design facilitates the information from the GT image and stabilizes the fine-tuning procedure. While [1, 2] only focus on the noise from pure noise.
>
> **Comment 2. Thorough user study**
>
> Thanks for your advice. We extend our original user preference study introduced in Appendix A.1.
>
> We categorize the alignment metric in the following aspects as in DSG1K: entities, attributes, relations and global. Entities contain the whole object and the part of the object. Attributes contain color, type, etc. Relations contain spatial and actions. Global contains other aspects like illumination, etc.
>
> 5 participants are asked to evaluate the 100 image pairs generated by SDXL and CoMat-SDXL. For each alignment metric, we set 1 as aligned and 0 as not aligned. We show the result in Table 2 in the author rebuttal PDF. Our method enhance the baseline model in all the metrics with the most significant improvement on entities.
>
> **Comment 3. Weak Contribution**
>
> Sorry for the confusion caused. We respectfully disagree the reviewer's assessment for the following reasons:
>
> 1. **No fundamental challenges solved**
>
>    First of all, we argue that the fundamental challenges of the text following ability span various domains. Both the reviewer's mentioned tasks (i.e., the counting-aware prompts and spatial-aware prompts) and other tasks like object existence and attribute binding are of the same importance. If the object concept in the prompt is not activated, the object will not even appear on the image, let alone the spatial, counting problem. Therefore, it is inappropriate to prior the counting-aware or spatial-aware tasks to the other tasks, and hence didiminish the importance of our work.
>
>    Besides, the results in Table 1 in the main paper reveals that our method successfully addresses the spatial-aware prompts and bring significant improvement to both SD1.5 and SDXL, where nearly 80% increase is witnessed for SD1.5. We also add the evaluation result for the counting-aware prompt (numeracy) introduced in [3]. The result is shown in Table 3 in author rebuttal PDF. Our training data does not contain prompts specifically designed to solve the counting-aware task, the improvement further testify the effectiveness of our method.
>
>    We will investigate how to specifically address the spatial-aware and counting-aware tasks in the future work.
>
> 2. **Focus on improving benchmark performance**
>
>    We provide both quantitative and qualitative evaluation for the zero-shot performance on long and complex text prompts in Table 2 and Fig. 11 in the main paper. The huge improvement proves our method's generalizability to various scenarios in the alignment tasks.
>
>    Besides, we provide visualization results in Fig. 12 to 14 in the main paper to prove our method improve the prompt following ability across different tasks. These results includes various challenges like spatial-aware prompts (the bottom of Fig. 13), attribute-aware prompts (the spider and the cabin in Fig. 12), complex prompts (the lighthouse and the man in Fig.12), etc.
>
>
>
> [1] Aligning Text-to-Image Diffusion Models with Reward Backpropagation, arXiv 2023
>
> [2] Directly Fine-Tuning Diffusion Models on Differentiable Rewards, ICLR 2024
>
> [3] T2I-CompBench++: An Enhanced and Comprehensive Benchmark for Compositional Text-to-image Generation

---

### Author Rebuttal · Authors · 2024-08-07

Overall author rebuttal:

We thank all reviewers for their thoughtful comments. We greatly appreciate all the reviewers' acknowledgment that our method is **effective and achieves excellent results**. We have added new evaluations and visualization in our author rebuttal PDF.

The main concerns raised by the reviewers revolve around technical details, experiment settings, and contributions.

We make a global response to the pixel-level guidance provided by the image-to-text model, i.e., the $\mathcal{L}\_{i2t}$ loss here:

As shown in Fig. 1 in the author rebuttal PDF, the image-to-text model offers pixel guidance by attending to the place for the wrong generation, e.g., the eraser, the street name, and the jersey. It also leaves the correct generation untouched, e.g., the yellow pencil, the number 21, and the silver tie.

For other individual comments or questions, we answer all of them under each review.

We are committed to incorporating these improvements and addressing all the raised concerns in our final draft.

---

### Decision · Program_Chairs · 2024-09-25

**Decision:**

Accept (poster)

**Comment:**

The paper received two borderline accepts, one weak accept, and one borderline reject. Reviewers acknowledged the effectiveness of the proposed method in addressing text-to-image misalignment issues. However, reviewers also found that the proposed method is quite complicated and may require a lot of computation. Besides, one reviewer commented that similar ideas have already been explored in two recent works; the difference is different design choices. After reading the paper, the reviews, and the rebuttal, AC considers that this study is still valuable for the research community to understand the proposed design choices and their effectiveness. The contributions were also acknowledged by three reviewers. The proposed method is indeed complicated, with many sub modules and different steps in the pipeline. Nevertheless, authors committed to release the codes to the public, which is a positive step, and it was also agreed by some reviewers. Therefore, AC would like to recommend the acceptance of this paper, and hope authors will keep their promises as in the rebuttal to improve the final version and release the code.